# Neuron-level Balance between Stability and Plasticity in Deep Reinforcement Learning

## Abstract

In contrast to the inherent ability of humans to continuously acquire new knowledge, modern deep reinforcement learning (DRL) agents generally encounter a significant challenge: the stability-plasticity dilemma, which refers to the trade-off between retaining existing skills (stability) and learning new knowledge (plasticity). In this study, we propose Neuron-level Balance between Stability and Plasticity (NBSP) to tackle this challenge, by taking inspiration from the observation that both stability and plasticity are integrally linked to the expressive capabilities of networks, which are primarily determined by the behavior of individual neurons. To the best of our knowledge, this is the first work that addresses both stability and plasticity loss simultaneously in DRL at the level of neurons. Specifically, NBSP first (1) defines and identifies RL skill neurons that are crucial for knowledge retention through a goal-oriented method, and then (2) introduces a stability-plasticity balancing mechanism by employing gradient masking and experience replay techniques targeting these neurons to preserve the encoded memory related to existing skills while enhancing the learning capabilities of other neurons. Experimental results on the Meta-World and Atari benchmarks demonstrate that NBSP significantly outperforms existing approaches in balancing stability and plasticity. Furthermore, our findings underscore the pivotal role of the critic within this context, providing valuable insights for future research.

## 1 Introduction

Deep reinforcement learning (DRL), which integrates the principles of both reinforcement learning (RL) and deep learning (DL), has shown exceptional capabilities across a range of complex scenarios, such as gaming (Mnih et al., 2013), robotic manipulation (Andrychowicz et al., 2020), and autonomous driving (Kiran et al., 2021). While most advancements in DRL focus on single-task scenarios, real-world applications often require agents to handle tasks sequentially, which remains under-explored in the research community. For example, a robot initially trained to open a window might subsequently require the ability to close it in response to rainy weather. Similarly, an autonomous vehicle designed to follow lanes under standard conditions must adapt to changing lanes to deal with obstacles. Naive solutions such as retraining an agent from scratch and developing a new agent for each task are either computationally expensive or impractical. To this end, a more efficient approach presents training the agent continually on the new task based on the knowledge acquired from the previous ones.

Despite the promise of the sequential RL learning approach, the **stability-plasticity dilemma** remains a fundamental and under-explored problem. Ideally, the agent must maintain its performance on the previously learned task, a characteristic referred to as **stability**(McCloskey & Cohen, 1989), while simultaneously adapting to the new task, known as **plasticity** (Carpenter & Grossberg, 1987). However, it has been revealed that emphasizing stability may hinder the ability of agents to learn new knowledge (Nikishin et al., 2022a; Abbas et al., 2023), whereas excessive plasticity can lead to catastrophic forgetting of previously acquired knowledge (Goodfellow et al., 2015; Atkinson et al., 2021), a challenge known as the **stability-plasticity dilemma** (eMermillod et al., 2013). Our preliminary experiments demonstrate that this dilemma persists in DRL when new tasks are learned directly, necessitating the development of a mechanism that effectively balances these two competing demands, which is the main focus of our work.

Existing methods to strike a balance between stability and plasticity generally fall into three categories, i.e. (1) **regularization-based methods** (Kumar et al., 2023), which apply penalties to parameter changes to mitigate forgetting while acquiring new knowledge; (2) **replay-based methods** (Ahn et al., 2024), which leverage past experiences to consolidate knowledge; and (3) **modularity-based methods** (Anand & Precup, 2024), which seek to decouple stability and plasticity or isolate different components for different tasks. Despite their contributions, these methods suffer from three key limitations: (1) They primarily operate at the network level. However, stability and plasticity attribute to the expressive capabilities, which are significantly influenced by the behavior of individual neurons. To this end, identifying and leveraging these skill neurons is critical yet under-researched. (2) These studies are primarily conducted within the framework of continual learning, thus overlooking the unique characteristics intrinsic to DRL. (3) These approaches could sometimes unnecessarily inflate model parameters, thereby introducing unwarranted complexity (Bai et al., 2023a).

Motivated by the aforementioned observations, in this work, we tackle these problems from the perspective of RL skill neurons, and propose **Neuron-level Balance between Stability and Plasticity (NBSP)**, a novel DRL framework that operates at the level of individual neurons. In particular, (1) we first introduce **RL skill neurons**, which are essential for retaining knowledge. While skill neurons have been extensively investigated and successfully exploited in various domains, such as pretrained language models (Wang et al., 2022) and neural machine translation (Bau et al., 2018), skill neurons are still much less explored in DRL. We bridge this research gap by proposing a goal-oriented strategy for identifying RL skill neurons. (2) We then apply **gradient masking** to these neurons to preserve the encoded memory related to previous skills while allowing other neurons to adapt to a new task. (3) Additionally, we incorporate **experience replay** to periodically revisit the previous task, preventing excessive drift from the original knowledge. Integrally, NBSP offers three key advantages compared with previous methods: (1) The neuron-level processing enables finer control and greater flexibility, addressing the stability-plasticity trade-off at the most fundamental level of the network. (2) The goal-oriented approach for identifying RL skill neurons is specifically tailored to DRL. (3) This framework is simple and parameter-free, avoiding complex network designs or additional parameters.

We conduct experiments on the **Meta-World** (Yu et al., 2020) and the **Atari** (Mnih et al., 2013) benchmarks to evaluate the effectiveness of NBSP. Our results demonstrate that NBSP achieves superior performance in balancing stability and plasticity compared with existing methods, which not only enables effective learning of the new task but also preserves knowledge from the previous task. Additionally, we analyze the DRL agents by dissecting the performance of the two critical modules, i.e., the actor and the critic, to assess their contributions in balancing stability and plasticity. Our findings reveal that (1) addressing both the actor and critic networks yields the best performance, and (2) the critic plays a more critical role in achieving this balance due to the differences in their inherent training mechanisms. In summary, our key contributions include:

- We are the first to introduce the concept of RL skill neurons which are essential for knowledge retention, and propose a goal-oriented strategy specifically tailored to DRL for identification.

- We tackle the stability-plasticity dilemma in DRL from the perspective of RL skill neurons, by employing gradient masking and experience replay on these neurons, eliminating requirements of complex network designs or additional parameters.

- We conduct extensive experiments on the Meta-World and Atari benchmarks to demonstrate the effectiveness of our method in balancing stability and plasticity. Furthermore, we highlight the crucial role of the critic in achieving this balance, shedding insights on future research.

## 2 RELATED WORK

**Balance between stability and plasticity**. Excessive focus on stability can impede the ability of agents to learn new knowledge, whereas too much plasticity can result in catastrophic forgetting of previously acquired knowledge (Jung et al., 2023). This issue, known as the plasticity-stability dilemma, was initially outlined by Carpenter & Grossberg (1988), highlighting the trade-off between preserving performance on the previously learned task and adapting to the new ones. Recent advancements have explored various strategies to address this dilemma. For instance, Cui et al. (2023) integrate a hyperbolic metric learning module (Hyper-Metric) into a distillation-based framework to manage the trade-off between retaining old knowledge and acquiring new insights. LODE

enhances balance by decoupling loss functions for new tasks, thereby effectively separating objectives (Liang & Li, 2024). PromptFusion tackles the stability-plasticity dilemma through a Stabilizer module to mitigate catastrophic forgetting and a Booster module to support concurrent learning of new knowledge (Chen et al., 2023). Moreover, Bai et al. (2023b) split the learning process into policy optimization and policy correction phases to strike an effective balance between stability and plasticity. While these methods primarily operate at the network level and are conducted within the framework of continual learning, our approach (1) delves into neuron-level research, which offers finer control and operates at the most fundamental level of networks, and (2) focuses on DRL by identifying RL skill neurons specifically tailored to DRL, beneficial for adapting to different tasks in DRL.

**Neuron-level research**. Recent studies indicate that not all neurons remain active across different contexts, and this neuron sparsity often correlates positively with task-specific performance (Xu et al., 2024). Concurrently, an increasing number of studies have focused on identifying skill neurons to interpret network behavior or address challenges, achieving notable successes. For instance, skill neurons in pre-trained Transformers, which exhibit high predictive value for task labels, have been leveraged to accelerate Transformers through network pruning and improve transferability (Wang et al., 2022). Sokar et al. (2023) explore dormant neurons in deep reinforcement learning, proposing a method to recycle these neurons throughout training. Dravid et al. (2023) identify Rosetta Neurons, which facilitate cross-class alignments, transformations without specialized training. Additionally, language-specific neurons within large language models have been shown to steer the output language by selectively activating or deactivating them (Tang et al., 2024). In the realm of safety alignment, safety neurons have been identified and analyzed from the perspective of mechanistic interpretability (Chen et al., 2024). Thought skill neurons successfully exploited in various domains, they are still much less explored in DRL. In contrast, we identify RL skill neurons that are specifically tailored to DRL, successfully balancing stability and plasticity in DRL.

# 3 BALANCE BETWEEN STABILITY AND PLASTICITY

## 3.1 THE PHENOMENON OF STABILITY AND PLASTICITY LOSSES

Extensive research has explored the loss of stability and plasticity in deep learning (Xie et al., 2021). This challenge is also prominent in DRL (Atkinson et al., 2021; Abbas et al., 2023), albeit with less exploration. We begin by conducting preliminary experiments to demonstrate the phenomenon in DRL, using four task pairs from the Meta-World benchmark (Yu et al., 2020). Then we evaluate the performance on the first task after sequentially learning the task pair and the learning progress on the second task after mastering the first one to indicate the stability and plasticity losses respectively.

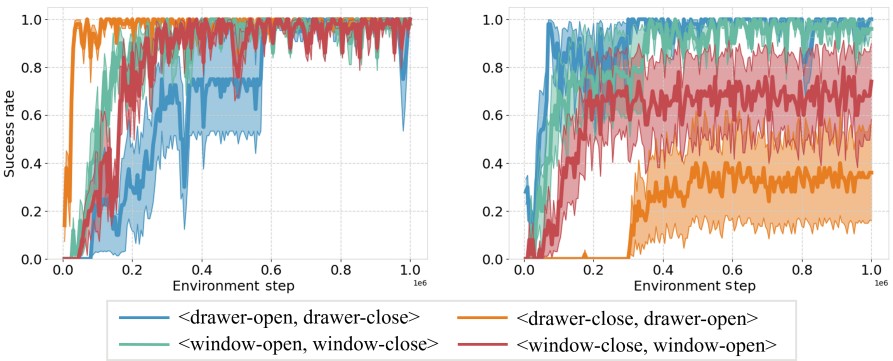

Figure 1: Training curves of sequential learning of task pairs. The left panel illustrates the training progress on the first task, while the right panel shows the progress on the second task. The solid line denotes the mean performance over five seeds, and the shaded area indicates the standard error.

**Stability loss**. As shown in Figure 1, the agent is able to effectively learn each task during most of the training process, even when transitioning between tasks. However, when the performance of the agent on the first task is evaluated after it has successfully learned the new task, a significant decline

Table 1: Evaluation performance on the first tasks. Sequential success rates denote the performance of the first tasks after sequentially learning the task pairs, while individual success rates indicate the performance after training solely on the first tasks.

| Task pairs | Individual success rates | Sequential success rates |
|---|---|---|
| \<drawer-open, drawer-close\> | $0.98 \pm 0.04$ | $0.00(\downarrow 0.98) \pm 0.00$ |
| \<drawer-close, drawer-open\> | $0.96 \pm 0.02$ | $0.62(\downarrow 0.34) \pm 0.20$ |
| \<window-open, window-close\> | $0.98 \pm 0.04$ | $0.00(\downarrow 0.98) \pm 0.00$ |
| \<window-close, window-open\> | $0.97 \pm 0.04$ | $0.09(\downarrow 0.88) \pm 0.08$ |

is observed, as indicated in Table 1. For example, the success rate for \<drawer-open, drawer-close\> pair drops dramatically from 0.98 to 0.00. This severe decline suggests that the agent has completely forgotten the previously acquired knowledge, leading to significant stability loss.

**Plasticity loss**. Beyond the stability loss, there are instances where the agent loses its ability to learn a new task after it has mastered a first one, which is evident in \<drawer-close, drawer-open\> pair. As illustrated in Figure 1 and Table 1, while the agent is able to learn drawer-open from scratch, it fails on this task after previously mastering drawer-close, indicating a clear loss of plasticity. In addition to the difficulty in learning the second task, the performance on the first task further deteriorates from 0.96 to 0.62, which suggests that the agent struggles to strike a balance between stability and plasticity.

## 3.2 IDENTIFYING RL SKILL NEURONS

In this study, we make a key observation that the stability and plasticity of a network are closely related to its expressive capabilities, which are significantly influenced by the behavior of individual neurons. As evidenced in Molchanov et al. (2022), neuron expression determines how information is propagated and processed within the neural network, directly affecting the learning and knowledge retention capabilities of the network. Therefore, understanding and controlling neuron behavior is at the most fundamental level for striking a balance between stability and plasticity. On the one hand, when neuron expression is stable and generalized, the network tends to exhibit high stability. On the other hand, strong plasticity can be achieved given neuron expression is flexible and adaptable.

Existing research has demonstrated the multifaceted capabilities of neurons such as the storage of factual knowledge (Dai et al., 2022), the association with specific languages (Tang et al., 2024), and encoding of safety information (Chen et al., 2024). These specialized neurons, often referred to as skill neurons, have been shown to significantly contribute to network performance (Wang et al., 2022). However, the potential of skill neurons in DRL remains largely under-explored. To bridge this research gap, we introduce **RL skill neurons**, which are essential for knowledge retention in DRL, and propose a goal-oriented method for the identification of these neurons. Unlike prior approaches that focus on the inputs triggering neuron activations (Bau et al., 2020; Gurnee & Tegmark, 2023), our method emphasizes their impact on achieving the ultimate goal, i.e. succeeding in finishing Meta-World tasks and attaining high scores in Atari games, by comparing the activation patterns between the neurons of agents that exhibit varying performance levels. In Section 4.2, we empirically show that the proposed goal-oriented method can better identify neurons that are truly encoding task-specific RL skills.

For a specific neuron $\mathcal{N}$, let $a(\mathcal{N}, t)$ represent its activation at step $t$. In fully connected layers, each output dimension corresponds to the activation of a specific neuron, while for convolution layers, the average value of each output channel represents the activation of a neuron. To evaluate the activation level of a neuron, we define the standard activation of neuron $\mathcal{N}$ as follows:

$$\overline{a}(\mathcal{N}) = \frac{1}{T} \sum_{t=1}^{T} a(\mathcal{N}, t), \qquad (1)$$

where $T$ represents the evaluation step. Similarly, we define the standard evaluation criterion of the network $\theta$ as follows:

$$\overline{q}_{\theta} = \frac{1}{T} \sum_{t=1}^{T} q_{\theta}(t), \qquad (2)$$

where $q_\theta(t)$ denotes the evaluation criterion function, which computes the performance of network $\theta$ at step $t$ and varies across different environments. The evaluation criterion measures the degree to which the agent approaches the goal, which depends on the specific goal of the task being addressed. For instance, in the Meta-World benchmark, the evaluation criterion is typically binary, indicating whether the agent successfully accomplishes the task, while in the Atari benchmark, the evaluation criterion is based on the return over an episode.

To distinguish the roles of neurons for different tasks, it is crucial to evaluate neuron activations specific to different goals. Intuitively, we can consider a neuron $\mathcal{N}$ to be positively contributing to the goal at step $t$ when its activation $a(\mathcal{N}, t)$ exceeds the standard activation $\overline{a}(\mathcal{N})$, i.e. $a(\mathcal{N}, t) > \overline{a}(\mathcal{N})$, while the evaluation criterion also exceeds the standard, i.e. $q(t) > \overline{q}_\theta$. To quantify this, we accumulate a batch of results and define the positive accuracy as follows:

$$Acc(\mathcal{N}) = \frac{\sum_{t=1}^{T} 1_{[1_{[a(\mathcal{N},t)>\overline{a}(\mathcal{N})]}=1_{[q(t)>\overline{q}_\theta]}]}}{T}. \tag{3}$$

Here, $1_{[condition]} \in \{0, 1\}$ denotes the indicator function, which returns 1 if and only if the specified condition is satisfied. While Eq. (3) assesses the positive contribution of neurons towards achieving the goal, where higher accuracy implies a greater significance of the neuron in producing better outcome, however, it overlooks neurons that exhibit a negative correlation with the goal but still carry valuable task-related knowledge. Specifically, when the activation of a neuron falls below its standard activation, the agent performs well conversely. To this end, we define a comprehensive score $Score(\mathcal{N})$ for the neuron that takes into account both positive and negative effects:

$$Score(\mathcal{N}) = max(Acc(\mathcal{N}), 1 - Acc(\mathcal{N})). \tag{4}$$

Subsequently, we rank all neurons in the network $\theta$, excluding those in the last layer, in descending order based on their scores. The neurons with the highest scores are identified as RL skill neurons, as they are instrumental in task-specific knowledge retention. And the number of RL skill neurons varies depending on the complexity of the task.

### 3.3 NEURON-LEVEL BALANCE BETWEEN STABILITY AND PLASTICITY

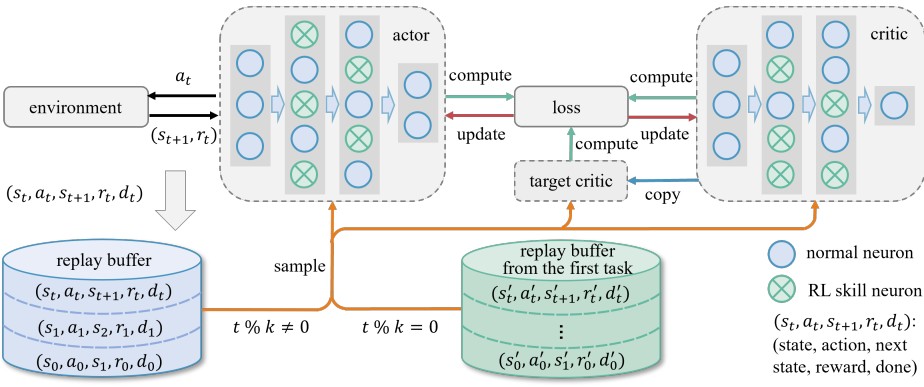

Figure 2: Framework of SAC with NBSP during the second task training. Green circles in the actor and critic networks represent RL skill neurons. The green replay buffer stores the experiences from the first task, and the blue stores from the current task. The experiences mainly contain the tuple of current state, action, next state, reward and whether the episode is done.

Building upon the concept of RL skill neurons, we propose a novel DRL framework — **Neuron-level Balance between Stability and Plasticity (NBSP)**. Unlike prior methods (Bai et al., 2023a; Kim et al., 2023), the framework proposed does not require complex network designs or additional parameters. To effectively preserve the encoded knowledge related to skill about the first task of RL skill neurons while bringing other neurons into full play to learn the knowledge of new task, NBSP employs a **gradient masking** technique. Before each update during the training process for the second task, we utilize the set of RL skill neurons to determine whether the gradient of a particular

neuron should be retained or zeroed out. This process can be formally expressed as follows:

$$\Delta W_{:,j}^{\prime(l)} = mask_j^{(l)} \cdot \Delta W_{:,j}^{(l)}, \tag{5}$$

where $\Delta W_{:,j}^{(l)}$ denotes the gradient with respect to the weight $W_{:,j}^{(l)}$ in the $l$-th layer of the network, and $j$ is the index of the output neuron in that layer. The term $mask_j^{(l)}$ is a binary value associated with the $j$-th neuron in the $l$-th layer. If the neuron is identified as a RL skill neuron, then $mask_j^{(l)} = 0$, otherwise it will be 1. The adjusted gradient $\Delta W_{:,j}^{\prime(l)}$ is obtained by modifying the original gradient based on the mask. Through gradient masking, RL skill neurons remain unaffected during each learning iteration while other neurons maintain great learning ability, striking a better balance between stability and plasticity, which is verified in Section 4.3.

To prevent excessive drift from the knowledge acquired from the first task, we incorporate **experience replay** technique to periodically sample the experiences stored in the replay buffer $D_{pre}$ from the first task at specific interval $k$, thereby further enhancing the stability of the DRL agents. The corresponding loss function reads:

$$\mathcal{L} = R(t) \cdot \mathbb{E}_{(s_t,a_t,s_{t+1},r_t,d_t)\sim D_{pre}}[L] + (1 - R(t)) \cdot \mathbb{E}_{(s_t,a_t,s_{t+1},r_t,d_t)\sim D}[L], \tag{6}$$

where $L$ denotes the original loss function, $R(t)$ is a binary function that evaluates to 1 if and only if the current step $t$ is at an interval. $D$ represents the replay buffer for the current task, and $(s_t, a_t, s_{t+1}, r_t, d_t)$ denotes the tuple of current state, action, next state, reward and whether the episode is done sampled from the replay buffer.

In this study, we implement NBSP within the Soft Actor-Critic (SAC) framework (Haarnoja et al., 2018). The overall architecture is illustrated in Figure 2. During the learning of the second task, the gradients of RL skill neurons within the actor and critic networks are masked to retain the knowledge acquired from the previous task. Additionally, we use two separate replay buffers for experience replay: one for storing the current experiences and the other for preserving experiences from the previous task. The agent then selectively samples from these buffers to update networks.

# 4 EXPERIMENT

In this section, we evaluate the performance of NBSP on the **Meta-World** and **Atari** benchmarks (Yu et al., 2020; Mnih et al., 2013). Additionally, we conduct ablation studies to (1) assess the effectiveness of the two primary components of NBSP, i.e. gradient masking and experience replay, and (2) investigate the effects on the two critical modules of DRL agents, i.e. the actor and the critic.

## 4.1 EXPERIMENTAL CONFIGURATION

**Benchmark**. We evaluate NBSP on task pairs from the Meta-World benchmark (Yu et al., 2020). The four tasks that we select to construct the task pairs are shown in Figure 3 as illustration. Based on the learning difficulty of each task, we design four task pairs with varying combinations of difficulty. More details on the Meta-world and the other Atari benchmarks are provided in Appendix C.2.

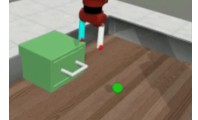 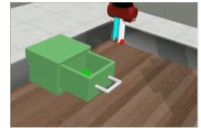

(a) drawer-open      (b) drawer-close

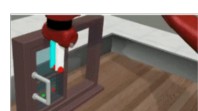

(c) window-open      (d) window-close

**Experiment setting**. For all experiments, we utilize the Soft Actor-Critic (SAC) (Haarnoja et al., 2018) algorithm implemented by CleanRL (Huang et al., 2022). Each agent is trained until reaching a predefined maximum step or stably mastering the task on the Meta-World benchmark. To ensure robustness, each experiment is repeated with five random seeds. Additional details can be found in Appendix C.3.

Figure 3: Tasks in the Meta-World benchmark.

**Metric**. The evaluation metric for the Meta-World benchmark is the success rate, as defined by Meta-World. Each task is evaluated based on an average success rate over 10 episodes. For the final performance evaluation, 200 episodes are sampled per task to calculate the average results.

For the Atari benchmark, the evaluation metric is the average return. In particular, we normalize the average returns using the random and human scores defined by Mnih et al. (2015) to obtain normalized scores for evaluating the final performance on the first game. Similarly, we sample 200 episodes to attain the average returns. Further details are available in Appendix C.4.

## 4.2 Results on the Meta-World benchmark

We compare NBSP with three state-of-the-art baselines: **(1) EWC** (Kirkpatrick et al., 2017), a classic regularization-based method for sequential task learning; **(2) ANCL** (Kim et al., 2023), a modularity-based approach that addresses the stability-plasticity dilemma with an auxiliary network; and **(3) Importance-based NBSP**, which is a variation of NBSP that replaces the RL skill neurons with important neurons (Paik et al., 2019). Detailed descriptions are provided in Appendix C.1. All approaches are evaluated on the same task pairs, with identical network architecture, to ensure a fair comparison. Part of the training results are illustrated in Figure 4, with additional results for different task pairs provided in Appendix C.5.2.

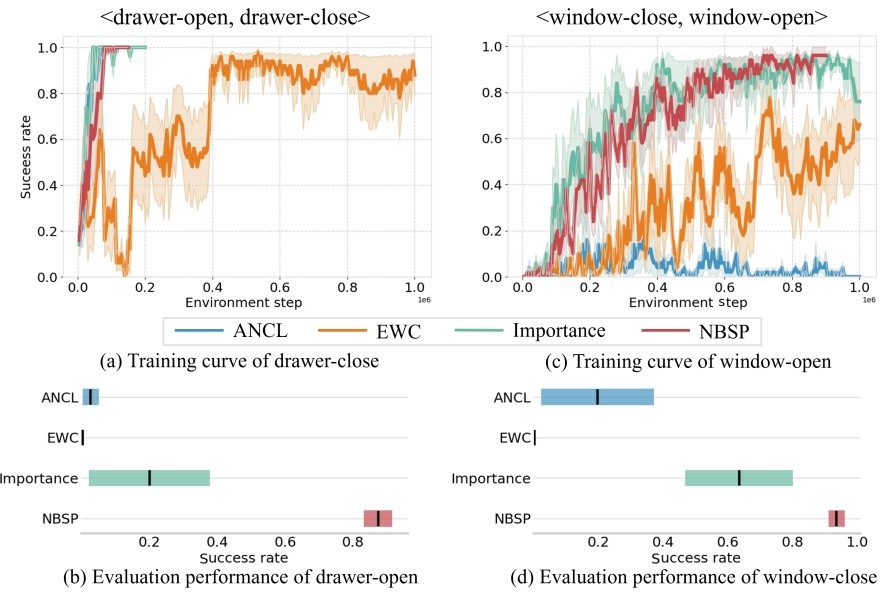

Figure 4: Comparison of NBSP on the Meta-World benchmark against other baselines. Figures (a) and (c) illustrate the training curves for the second task, with the solid line representing the average performance across five random seeds and the shaded region indicating the standard error. Figures (b) and (d) show the performance on the first task after sequentially learning the task pair, where the bar length reflects the standard error and the black vertical line denotes the actual average value. (Training terminates upon reaching stable optimal performance or maximum steps for each task.)

As illustrated in Figure 4, *NBSP achieves an optimal balance between stability and plasticity*. For all four task pairs evaluated, the average success rates approach 1.0 as training progresses on the second task, indicating that NBSP effectively adapts to the second task (plasticity). Furthermore, the agent maintains a high success rate, closing to 1.0 on the first task after learning the task pair, robustly retaining the prior knowledge (stability).

In contrast, (1) EWC struggles to balance stability and plasticity simultaneously. For instance, in <drawer-open, drawer-close> pair, EWC excels in learning the drawer-close task but forgets the drawer-open task, resulting in frequent failures. Similarly, in <drawer-close, drawer-open> pair shown in Appendix C.5.2, while EWC performs well on the first task, it fails to learn the second task due to limitations in network parameters. (2) ANCL, though effective at balancing stability and plasticity in classification tasks, performs poorly in DRL. For example, in <window-close, window-open> pair, ANCL struggles to maintain both stability and plasticity, exhibiting deficiencies in both aspects. (3) Importance-based NBSP, a variant of NBSP that masks important neurons rather than RL skill neurons, shows effectiveness in a specific task pair — <window-close, window-open>

pair. However, it also faces challenges in balancing stability and plasticity in other task pairs, as evidenced by a significant decline in knowledge retention for the drawer-open task after mastering the drawer-close task. *The comparison between Importance-based NBSP and our NBSP based on RL skill neurons highlights the prominent role of RL skill neurons, validating that our goal-oriented method does identify the neurons that essential for knowledge retention.*

## 4.3 ABLATION STUDY

In this section, we further evaluate the effectiveness of (1) the two primary components of NBSP: the **gradient masking** technique and **experience replay** technique, and (2) the two critical modules of DRL: the **actor** and the **critic**.

**Gradient masking and experience replay**. We design three settings for this ablation study: **(1) Base**: training the task pair directly without any additional techniques. **(2) Replay-Only**: training the task pair with only the experience replay technique. **(3) NBSP**: training the task pair with both gradient masking and experience replay techniques.

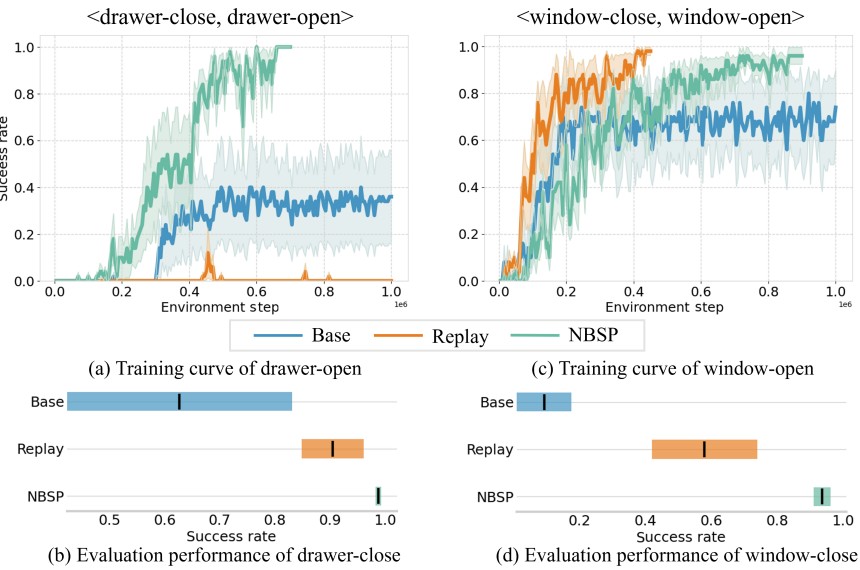

(a) Training curve of drawer-open      (c) Training curve of window-open

(b) Evaluation performance of drawer-close      (d) Evaluation performance of window-close

Figure 5: Results of the two primary components of NBSP on the Meta-World benchmark. Figures (a) and (c) illustrate the training curves for the second task, while Figures (b) and (d) show the performance on the first task after sequentially learning the task pair. (Training terminates upon stably optimal performance or maximum steps for each task.)

Results of two task pairs are illustrated in Figure 5. (1) It can be observed that training the task pairs without any technique results in severe knowledge forgetting and poorer ability to learn new knowledge, which is discussed in Section 3.1. (2) The experience replay technique is beneficial for retaining knowledge to some extent, as seen in <drawer-close, drawer-open> and <window-close, window-open>. These results empirically demonstrate that the experience replay technique helps prevent excessive drift from the original knowledge, thereby enhancing the stability of the DRL agents. (3) Notably, **NBSP** achieves superior performance over both **Base** and **Replay-Only** settings. In particular, NBSP not only excels in retaining knowledge encoded by RL skill neurons but also adapts to the second task with the learning capabilities of other neurons. *These findings highlight the effectiveness of both the gradient masking and experience replay techniques in achieving an optimal balance between stability and plasticity, especially the prominent role of gradient masking technique for RL skill neurons.* Additional results of different task pairs are provided in Appendix C.5.3.

**Actor and critic**. To get a deeper understanding of the individual roles of the actor and critic in DRL agents, we evaluate two variations of our method: **(1) NBSP-Actor**: applying NBSP exclusively to the parameters of the actor network; **(2) NBSP-Critic**: applying NBSP exclusively to the parameters

of the critic network. Main results are presented in Figure 6, and we provide additional results of other task pairs in Appendix C.5.4.

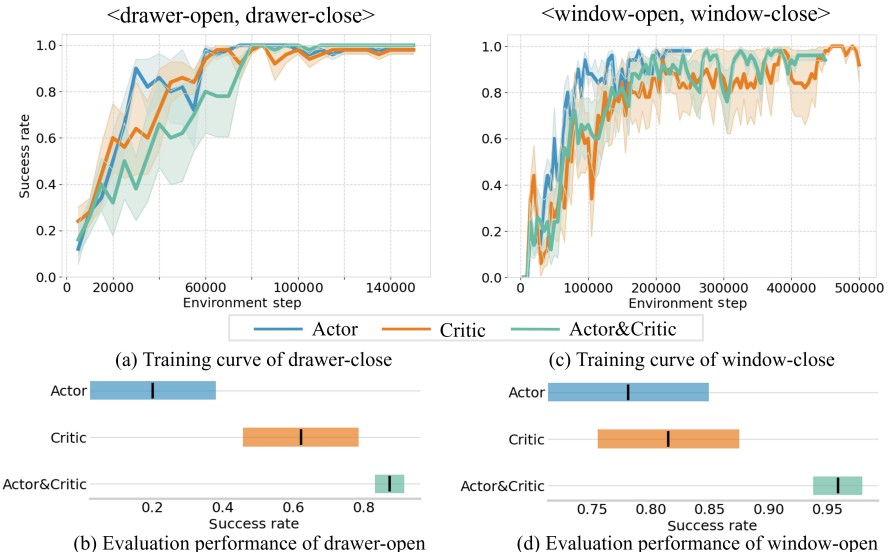

(a) Training curve of drawer-close

(b) Evaluation performance of drawer-open

(c) Training curve of window-close

(d) Evaluation performance of window-open

Figure 6: Performance comparison of NBSP with two variations . Figures (a) and (c) illustrate the training curves for the second task, while Figures (b) and (d) show the performance on the first task after sequentially learning the task pair. (Training terminates upon stably optimal performance or maximum steps for each task.)

As illustrated in Figure 6, for most task pairs, such as <drawer-open, drawer-close> and <window-open, window-close>, both NBSP-Actor and NBSP-Critic achieve substantial improvements in learning the second task while retaining knowledge from the first one. Notably, NBSP-Critic performs better in retaining knowledge compared to NBSP-Actor. Despite these improvements, neither NBSP-Actor nor NBSP-Critic achieves the same level of performance as NBSP, which applies the techniques to both the actor and critic networks simultaneously.

*Our findings indicate that both the actor and critic networks are essential for striking an optimal balance between stability and plasticity. Notably, the critic proves to be the more critical module in balancing this trade-off*, which aligns with the insight from Ma et al. (2024) that plasticity loss in the critic serves as the principal bottleneck impeding efficient training in DRL. *We further investigate this phenomenon by dissecting the inherent training mechanisms of actor-critic RL methods, and draw the following key observations*: (1) Updates to the actor network are guided by feedback from the critic. Consequently, even if the RL skill neurons in the actor are masked, they remain influenced by the critic, which may gradually adapt to the new task at the expense of retaining prior knowledge; (2) In contrast, applying NBSP to the critic network indirectly constrains the actor as well; and (3) The update process of the critic network is recursive, with its target network updated via an exponential moving average, enabling it to preserve knowledge from the previous task while integrating new skills. Therefore, NBSP achieves better performance on the critic than on the actor. This demonstrates the distinct roles of the actor and critic networks in balancing stability and plasticity, providing valuable insights for future research in this field.

## 4.4 EXPERIMENTS ON THE ATARI BENCHMARK

We further evaluate on the Atari benchmark to investigate the generalization ability of NBSP. Unlike the continuous action space in Meta-World, Atari games are characterized by discrete action spaces, and episode returns are used to assess the performance of each game. We conduct experiments on four game pairs, including two irrelevant and two relevant game pairs, and train each with five different seeds. The results of two game pairs are presented in Figure 7, with additional results of other two game pairs provided in Appendix C.5.5.

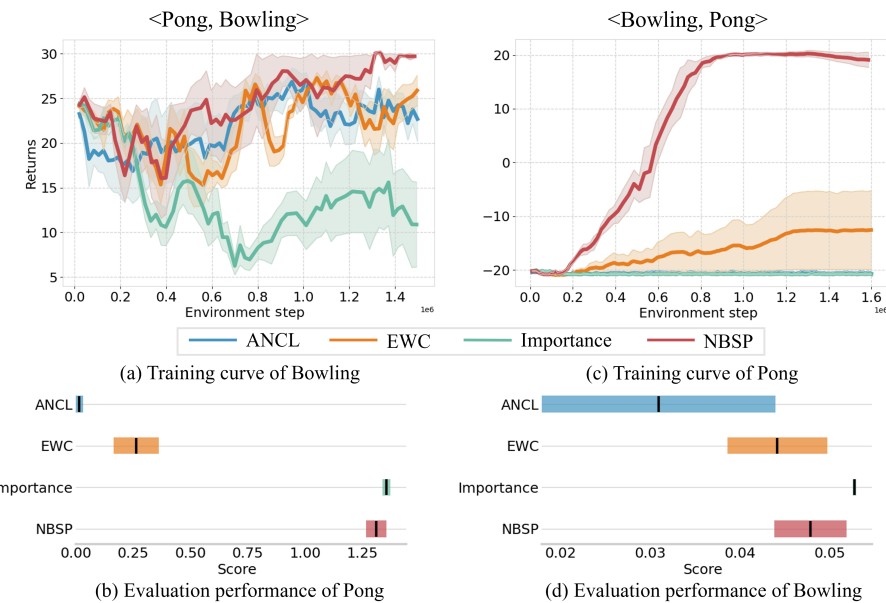

Figure 7: Results of NBSP on the Atari benchmark. Figures (a) and (c) illustrate the training curves for the second game, while Figures (b) and (d) show the performance on the first game after sequentially learning the game pair.

As shown in Figure 7, NBSP successfully strikes a balance between stability and plasticity on the Atari benchmark. Compared to other methods, NBSP excels in learning the second game while preserving knowledge from the first game. In contrast, ANCL exhibits significant challenges, often losing either plasticity, which hinders the learning of the new game, or stability, resulting in forgetting previously acquired knowledge. Similarly, EWC faces difficulties in balancing both stability and plasticity, though it performs better than ANCL. And Importance-based NBSP demonstrates improved knowledge retention from the first game but struggles to effectively learn the second game. *Overall, NBSP achieves a balance between stability and plasticity on both Meta-World and Atari benchmarks, demonstrating its generalization ability.*

## 5 CONCLUSION AND FUTURE WORK

**Conclusion**. This work addresses the fundamental issue of the stability-plasticity dilemma in DRL. We begin by illustrating the challenges DRL agents face in retaining knowledge from a prior task while effectively learning a new one. To tackle this problem, we introduce the concept of RL skill neurons by identifying neurons that significantly contribute to knowledge retention, building upon which we then propose the Neuron-level Balance between Stability and Plasticity framework, by employing gradient masking and experience replay techniques on RL skill neurons. Experimental results on the Meta-World and Atari benchmarks demonstrate that NBSP significantly outperforms existing methods in managing the stability-plasticity trade-off. Furthermore, comparative studies of NBSP with its variations reveal that the critic network plays a more crucial role in achieving this balance than the actor network, which is consistent with their respective training mechanisms, shedding valuable insights on future research in this field.

**Future work**. This work defines RL skill neurons, effectively balancing stability and plasticity in DRL. Future research could explore model distillation by pruning neurons apart from RL skill neurons for more compact models, and bias control by manipulating RL skill neuron activations for targeted behaviors. The NBSP method could also extend to other learning paradigms, such as supervised and unsupervised learning, to tackle similar challenges. We plan to investigate these directions further across diverse domains.

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

## A    EXTENDED RELATED WORK

**Plasticity**. In deep reinforcement learning, plasticity refers to the agent's ability to learn and adapt to the new task. A common risk in this process is overfitting to previous experiences, which can hinder subsequent learning, a phenomenon known as plasticity loss (Lyle et al., 2024). This issue is not unique to reinforcement learning but is also prevalent in continual learning, prompting the development of various strategies to preserve plasticity (Dohare et al., 2024). For instance, Nikishin et al. (2022b) propose a simple yet broadly applicable mechanism that addresses plasticity loss by periodically resetting a portion of the agent's network. Similarly, Nikishin et al. (2024) introduce plasticity injection, a minimalist intervention that enhances network plasticity without increasing the number of trainable parameters or introducing bias into predictions. Another approach, Neuron-level Plasticity Control (NPC), seeks to preserve existing knowledge by adjusting the learning rates of individual neurons (Paik et al., 2019). While these methods effectively enhance plasticity, they may still face the challenge of catastrophic forgetting when the agent attempts to learn a new task based on previously acquired knowledge.

**Stability**. Stability refers to the ability to retain previously acquired knowledge. Excessive plasticity can result in increased forgetting of earlier information, a phenomenon known as stability loss (Kemker et al., 2018). Current research primarily focuses on mitigating catastrophic forgetting (French, 1999) to preserve stability. For example, DR-Tune utilizes distribution regularization by enforcing the downstream task head to reduce classification errors on the pre-trained feature distribution, thereby preventing overfitting (Zhou et al., 2023). Similarly, the Adaptive Logit Regularizer (ALI) enhances the model's ability to learn new categories while maintaining knowledge of previous ones (Oh et al., 2022). Additionally, some approaches incorporate synaptic models designed to mitigate catastrophic forgetting across multiple timescales (Kaplanis et al., 2018). Although these methods are effective in preserving stability by constraining changes to the model, they often compromise plasticity. In contrast, our proposed method strikes a balance, maintaining stability while still enabling effective learning of the new task.

## B    PRELIMINARIES

### B.1    MARKOV DECISION PROCESS (MDP)

A Markov Decision Process(MDP) is a framework used to describe a problem involving learning from actions to achieve a goal. Almost all reinforcement learning problems can be characterized as a Markov Decision Process. Each MDP is defined by a tuple $< S, A, P, R, \gamma >$, where $S$ and $A$ represent state and action spaces respectively. The transition dynamics of the MDP are defined by the function $P : S \times A \times S \to [0, 1]$, which represents the probability of transitioning from a give state $s$ with action $a$ to state $s'$. The reward function is represented by $R : S \times A \times S \to \mathbb{R}$, and $\gamma \in (0, 1)$ is the discount factor. At each time step $t$, an agent observes the state of the environment, denoted as $s_t$, and selects an action $a_t$ according to a policy $\pi(a|s)$. One time step later, the agent receives a numerical reward $r_{t+1}$ and transitions to a new state $s_{t+1}$. In the simplest case, the return is the sum of the rewards when the agent–environment interaction naturally breaks into subsequences, which we refer to episodes (Sutton, 2018).

## B.2 Soft Actor-Critic (SAC)

Soft Actor-Critic (SAC) is an off-policy actor-critic deep reinforcement learning algorithm that leverages maximum entropy to promote exploration. This work employs SAC to train a policy that effectively balances stability and plasticity , chosen for its sample efficiency, excellent performance, and robust stability. In this framework, the actor aims to maximize both the expected reward and the entropy of the policy. The parameters $\phi$ of the actor are optimized by minimizing the following loss function:

$$J_\pi(\phi) = E_{s_t \sim D, a_t \sim \pi_\phi}[\alpha log \pi_\phi(a_t|s_t) - Q_\theta(s_t, a_t)],$$

where $D$ is the replay buffer, $\alpha$ is the temperature parameter controlling the trade-off between exploration and exploitation, $\theta$ denotes the parameters of the critic network, $\pi_\phi$ represents the policy learned by the actor $\phi$ , and $Q_\theta$ denotes the Q-value estimated by the critic $\theta$. The critic network is trained to minimize the squared residual error:

$$J_Q(\theta) = E_{(s_t, a_t, s_{t+1}) \sim D}[\frac{1}{2}(Q_\theta(s_t, a_t) - r_t - \gamma\hat{V}(s_{t+1})],$$

$$\hat{V}(s_t) = E_{a_t \sim \pi_\phi}[Q_\theta(s_t, a_t) - \alpha log \pi_\phi(a_t|s_t)],$$

where $\gamma$ represents the discount factor.

## B.3 Neuron

In neural networks, various components, such as blocks and layers, play distinct roles. Here, we define a neuron as a single output dimension from a layer. For example, in a fully connected layer, each output dimension corresponds to a neuron. Similarly, in a convolutional layer, each output channel represents a neuron. Furthermore, following the terminology used by Sajjad et al. (2022), we classify neurons that encapsulate a single concept as focused neurons, while a group of neurons collectively representing a concept are termed group neurons.

## C EXPERIMENTS

### C.1 BASELINES

**EWC**: Elastic Weight Consolidation (EWC) addresses the challenge of catastrophic forgetting by allowing neural networks to retain proficiency in previously learned tasks even after a long hiatus. It achieves this by selectively slowing down learning for weights that are crucial for retaining knowledge of these tasks. This approach has demonstrated excellent performance in sequentially solving a series of classification tasks, such as those in the MNIST handwritten digit dataset, and in learning several Atari 2600 games sequentially.

**ANCL**: Auxiliary Network Continual Learning (ANCL) is an innovative approach that incorporates an auxiliary network to enhance plasticity within a model that primarily emphasizes stability. Specifically, this framework introduces a regularizer that effectively balances plasticity and stability, achieving superior performance over strong baselines in both task-incremental and class-incremental learning scenarios.

**Importance-based NBSP**: To assess the significance of individual neuron, Paik et al. (2019) employ a criterion based on Taylor expansion, a technique commonly used in network pruning. They normalize this criterion and apply exponential moving averages (EMA) to retain the importance derived from previous tasks. Then Important neurons are consolidated by applying lower learning rates, thereby preserving knowledge from earlier tasks while continuing to learn new ones. We replace the RL skill neurons with important neurons to create the Importance-based NBSP.

### C.2 BENCHMARK

**Meta-World**. Meta-World is an open-source benchmark for meta-reinforcement learning and multitask learning, comprising 50 distinct robotic manipulation tasks (Yu et al., 2020).

All tasks are executed by a simulated Sawyer robot, with the action space defined as a 2-tuple: the change in the 3D position of the end-effector, followed by a normalized torque applied to the gripper fingers.

The observation space has a consistent dimensionality of 39, although different dimensions correspond to various aspects of each task. Typically, the observation space is represented as a 6-tuple, including the 3D Cartesian position of the end-effector, a normalized measure of the gripper's openness, the 3D position and the quaternion of the first object, the 3D position and quaternion of the second object, all previous measurements within the environment, and the 3D position of the goal.

The reward function for all tasks is structured and multi-component, aiding in effective policy learning for each task component. With this design, the reward functions maintain a similar magnitudes across tasks, generally ranging between 0 and 10. The descriptions of the four tasks used in our experiments are listed below.

- **drawer-open**: Open a drawer, with randomized drawer positions.
- **drawer-close**: Push and close a drawer, with randomized drawer positions.
- **window-open**: Push and open a window, with randomized window positions.
- **window-close**: Push and close a window, with randomized window positions.

Based on learning difficulty, which is assessed by the number of steps required to achieve stable performance, each task is classified as either easy or difficult. For example, drawer-open is considered a difficult task, whereas the other three tasks are categorized as easy. Consequently, our four experimental task pairs represent various combinations of difficulty:

- **drawer-open and drawer-close**: A difficult-to-easy combination.
- **drawer-close and drawer-open**: An easy-to-difficult combination.
- **window-open and window-close**: An easy-to-easy combination.
- **window-close and window-open**: An easy-to-easy combination.

**Atari**. Atari environments are simulated using the Arcade Learning Environment (ALE) (Bellemare et al., 2013) via the Stella emulator.

Each environment utilizes a subset of the full action space, which includes actions like NOOP, FIRE, UP, RIGHT, LEFT, DOWN, UPRIGHT, UPLEFT, DOWNRIGHT, DOWNLEFT, UPFIRE, RIGHTFIRE, LEFTFIRE, DOWNFIRE, UPRIGHTFIRE, UPLEFTFIRE, DOWNRIGHTFIRE, and DOWNLEFTFIRE. By default, most environments employ only a smaller subset of these actions, excluding those that have no effect on gameplay.

Observations in Atari environments are RGB images displayed to human players, with $obs\_type = "rgb"$, corresponding to an observation space defined as $Box(0, 255, (210, 160, 3), np.uint8)$.

The specific reward dynamics vary depending on the environment and are typically detailed in the game's manual.

The descriptions of the six games used in our experiments are listed below (Foundation, 2024), and the appearance of these games is shown in Figure 8.

- **Bowling**: The goal is to score as many points as possible in a 10-frame game. Each frame allows up to two tries. Knocking down all pins on the first try is called a "strike", while doing so on the second try is a "spare". Failing to knock down all pins in two attempts results in an "open" frame.
- **Pong**: You control the right paddle and compete against the computer-controlled left paddle. The objective is to deflect the ball away from your goal and into the opponent's goal.
- **DemonAttack**: Set on the ice planet of Krybor, you face waves of demons. Points are accumulated by destroying them. You start with three bunkers, and can increase this number (up to six) by avoiding enemy attacks. Surviving a wave without being hit grants an additional bunker. Once all bunkers are destroyed, the next hit ends the game.
- **Carnival**: A shoot game where you control a gun that moves horizontally to shoot targets moving across the screen. Ammunition is limited, and chickens can steal bullets if not hit in time.

- **BankHeist**: You play as a bank robber trying to rob as many banks as possible while avoiding the police in maze-like cities. You can destroy police cars using dynamite and refill your gas tank by entering new cities. Lives are lost if you run out of gas, are caught by the police, or run over your own dynamite.

- **Alien**: You are trapped in a maze-like spaceship with three aliens. Your goal is to destroy their eggs scattered throughout the ship while avoiding the aliens. You have a flamethrower to fend them off and can occasionally collect a power-up (pulsar) that temporarily enables you to kill aliens.

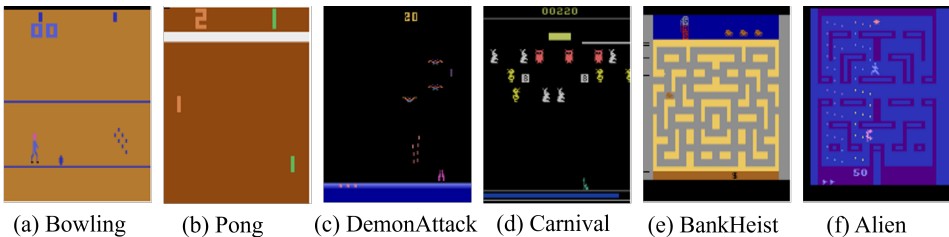

(a) Bowling      (b) Pong      (c) DemonAttack    (d) Carnival      (e) BankHeist      (f) Alien

Figure 8: Games in the Atari benchmark used in our experiments.

In our experiments, we examine four game pairs. **Bowling and Pong**, as well as **Pong and Bowling**, represent irrelevant game pairs, as they share little in common. Conversely, **DemonAttack and Carnival** are both shooting games involving avoiding attacks, while **BankHeist and Alien** both occur in maze-like environment, making them relevant game pairs. Thus, our experiments include two irrelevant and two relevant game pairs.

## C.3 DETAILED EXPERIMENT SETTINGS

For all experiments, we utilize the open-source PyTorch implementation of Soft Actor-Critic (SAC) provided by CleanRL (Huang et al., 2022) on a single RTX2080Ti GPU. CleanRL is a Deep Reinforcement Learning library that offers high-quality, single-file implementations with research-friendly features. The code is both clean and straightforward, and we adhere to the configurations provided by CleanRL. During training, we employ an $\epsilon$-greedy exploration policy at the start, setting $\epsilon = 1$ for the first $10^4$ time steps to promote exploration. The environment is wrapped using Gym wrappers to facilitate experimentation. For the Meta-World benchmark, we utilize the RecordEpisodeStatistics wrapper to gather episode statistics. For the Atari benchmark, in addition to RecordEpisodeStatistics, we preprocess the $210 \times 160$ pixel images by downsampling them to $84 \times 84$ using bilinear interpolation, converting the RGB images to the YUV format, and using only the grayscale channel. Additionally, we set a maximum limit on the number of noop and skip steps to standardize the exploration.

Regarding network architecture, we use the same actor and critic networks for all tasks within the same benchmark to ensure consistency. For the Meta-World benchmark, we employ a neural network comprising three fully connected layers with 256 hidden units each. For the Atari benchmark, we use a convolutional neural network (CNN) with three convolutional layers featuring 32, 64, and 64 channels, respectively, followed by two fully connected layers with 512 hidden units.

To reduce randomness and enhance the reliability of our results, we train each agent using five random seeds. Additional hyper-parameters for the SAC algorithm applied in the Meta-World and Atari benchmarks are detailed in Table 2.

## C.4 METRICS

For the Meta-World benchmark, performance is evaluated using the average success rate, defined as follows:

$$R = \frac{1}{|E|} \sum_{e \in E} r(e), \tag{7}$$

Table 2: Hyper-parameters of SAC in our experiments.

| Parameters | Values for Meta-World | Values for Atari |
|---|---|---|
| Initial collect steps | 10000 | 20000 |
| Discount factor | 0.99 | 0.99 |
| Training environment steps | $10^6$ | $1.5 \times 10^6$ 3, $\times 10^6$ |
| Testing environment steps | $10^5$ | $10^5$ |
| Replay buffer size | $10^6$ | $2 \times 10^5$ |
| Updates per environment step (Replay Ratio) | 2 | 4 |
| Target network update period | 1 | 8000 |
| Target smoothing coefficient | 0.005 | 1 |
| Optimizer | Adam | Adam |
| Policy learning rate | $3 \times 10^{-4}$ | $10^{-4}$ |
| Q-value learning rate | $10^{-3}$ | $10^{-4}$ |
| Minibatch size | 256 | 64 |
| Alpha | 0.2 | 0.2 |
| Autotune | True | True |
| Average environment steps of success rate | 10 | - |
| Stable threshold to finish training | 0.9 | - |
| Replay interval | 10 | 10 |
| No-op max | - | 30 |
| Target entropy scale | - | 0.89 |

where $E$ represents a batch of episodes, $|E|$ is the total number of episodes, and $r(e)$ indicates the success or failure of episode $e$. Specifically, $r(e) = 1$ if the episode is successful, and $r(e) = 0$ otherwise. During training, the average success rate is computed over 10 episodes, while for evaluating the first task after training, it is summarized over 200 episodes.

For the Atari benchmark, performance is assessed by the average return of each episode. To evaluate the final performance on the first task after training on task pairs, we normalize the score for each game to obtain summary statistics across games, as follows:

$$R = \frac{r_{agent} - r_{random}}{r_{human} - r_{random}}, \tag{8}$$

where $r_{agent}$ represents the average return evaluated over $10^5$ steps, the random score $r_{random}$ and human score $r_{human}$ are consistent with those used by Mnih et al. (2015), as detailed in Table 3.

Table 3: Normalization scores of Atari games.

| games | $r_{random}$ | $r_{human}$ |
|---|---|---|
| Bowling | 23.1 | 154.8 |
| Pong | -20.7 | 9.3 |
| DemonAttack | 152.1 | 3401.3 |
| BankHeist | 14.2 | 734.4 |

## C.5 ADDITIONAL RESULTS

### C.5.1 TRAINING CURVES OF THE FIRST TASK

In all experiments, the learning process for the first task remains consistent across different methods. Figure 9 illustrates the learning curves for four tasks in the Meta-World benchmark, while Figure 10 displays the learning curves for four games in the Atari benchmark.

### C.5.2 ADDITIONAL RESULTS ON THE META-WORLD BENCHMARK

The results of NBSP and other baselines on <drawer-close, drawer-open> pair, as well as <window-open, window-close> pair, are presented in Figure 11. As demonstrated, NBSP con-

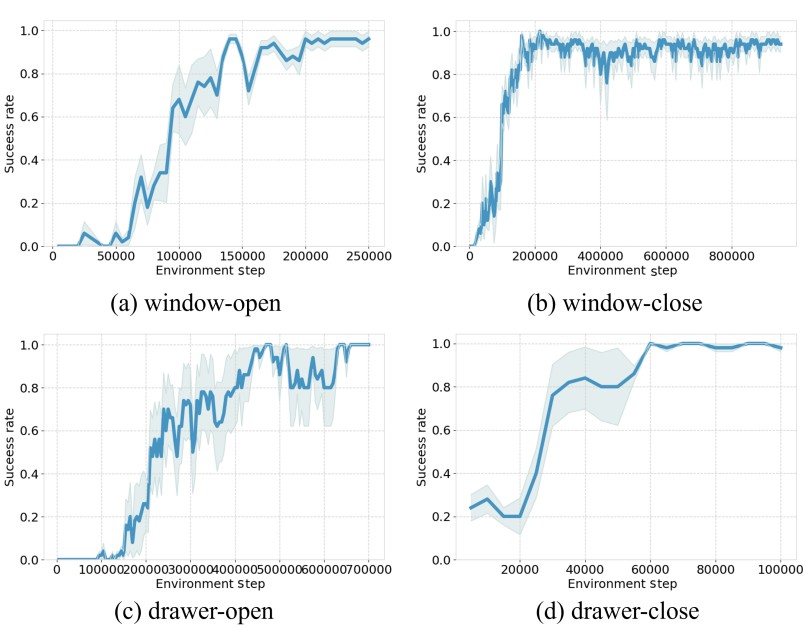

Figure 9: Training curves for the four individual tasks used in the experiments, including window-open, window-close, drawer-open and drawer-close.

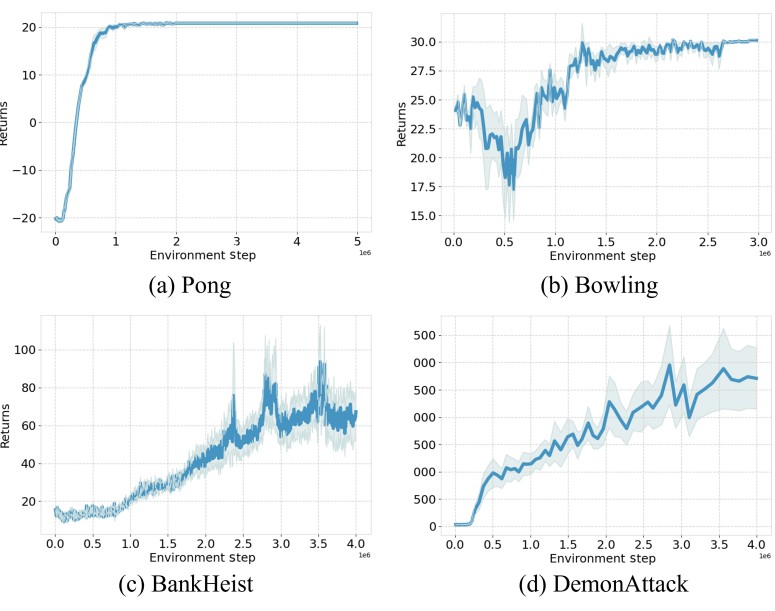

Figure 10: Training curves of the four individual games used in the experiments, including Pong, Bowling, BankHeist and DemonAttack.

sistently outperforms the other baselines, excelling both in acquiring new task knowledge and preserving previously learned knowledge. In contrast, the other baselines struggle to achieve a balance between stability and plasticity. These results further verify the effectiveness of NBSP on the Meta-World benchmark.

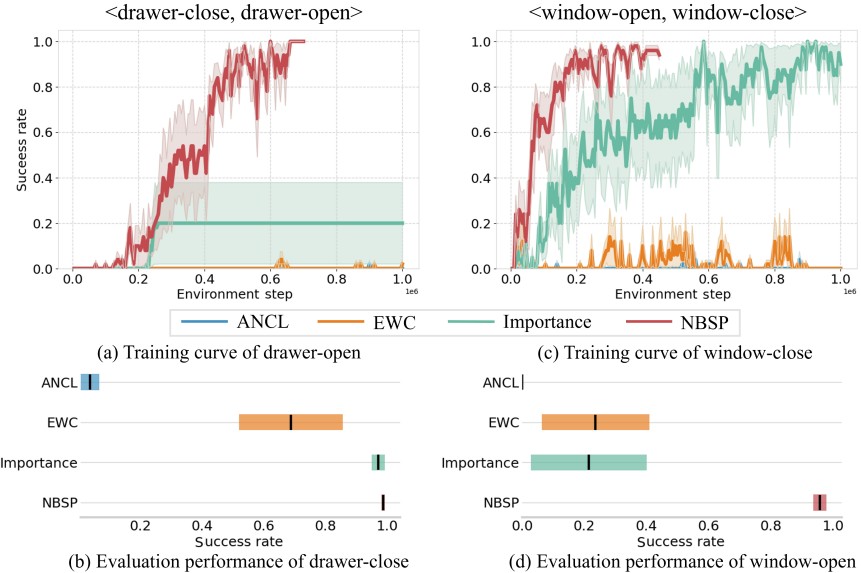

Figure 11: Additional results of NBSP on the Meta-World benchmark compared to other baselines. Figures (a) and (c) illustrate the training curves for the second task, while Figures (b) and (d) show the performance on the first task after sequentially learning the task pair. (Training terminates upon stably optimal performance or maximum steps for each task.)

### C.5.3 ADDITIONAL RESULTS OF THE TWO PRIMARY COMPONENTS OF NBSP

For <drawer-open, drawer-close> pair, as well as <window-open, window-close> pair, performance using the experience replay technique is superior to that without any technique, while NBSP significantly outperforms both, as illustrated in Figure 12. This highlights the effectiveness of both experience replay and gradient masking techniques in balancing stability and plasticity, particularly in retaining prior knowledge, with the gradient masking technique based on RL skill neurons making a notable contribution.

### C.5.4 ADDITIONAL RESULTS OF THE TWO CRITICAL MODULES OF DRL

Additional results showcasing variations of NBSP on the Meta-World benchmark are presented in Figure 13. Overall, NBSP demonstrates superior performance compared to its two variants — NBSP-Actor and NBSP-Critic. Among the two, NBSP-Critic achieves a better balance between stability and plasticity compared to NBSP-Actor.

### C.5.5 ADDITIONAL RESULTS ON THE ATARI BENCHMARK

The additional results for two game pairs in the Atari benchmark are shown in Figure 14. In <BankHeist, Alien> pair, NBSP exhibits exceptional plasticity, achieving a learning ability comparable to EWC and significantly surpassing ANCL and Importance-based NBSP. Moreover, NBSP retains strong performance on the first game even after sequentially learning the game pair, outperforming the other three baselines. Among these, EWC maintains stong plasticity but sacrifices stability, while Importance-based NBSP preserves some stability at the expense of plasticity. ANCL, however, underperforms in both aspects. For <DemonAttack, Carnival> pair, NBSP effectively balances stability and plasticity. Although it does not achieve the highest scores in terms of learning

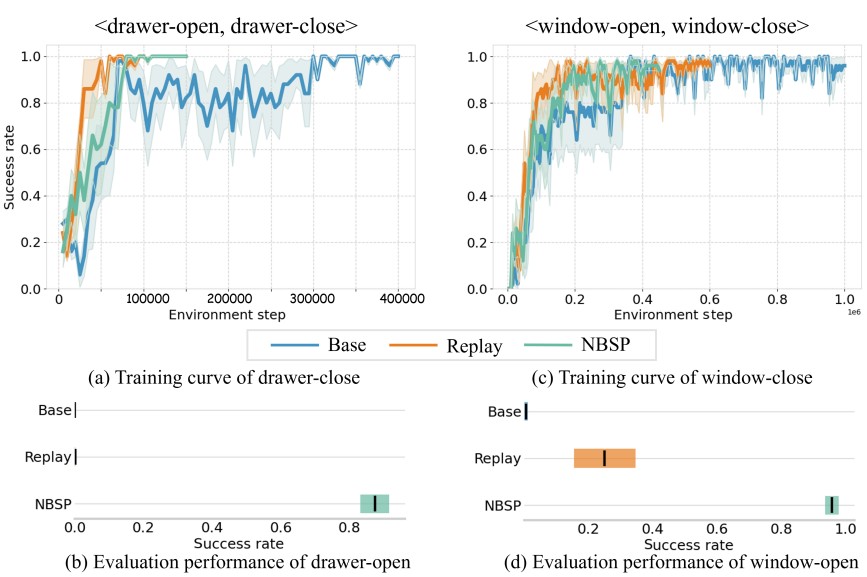

Figure 12: Additional results of ablation study on the Meta-World benchmark. Figures (a) and (c) illustrate the training curves for the second task, while Figures (b) and (d) show the performance on the first task after sequentially learning the task pair. (Training terminates upon stably optimal performance or maximum steps for each task.)

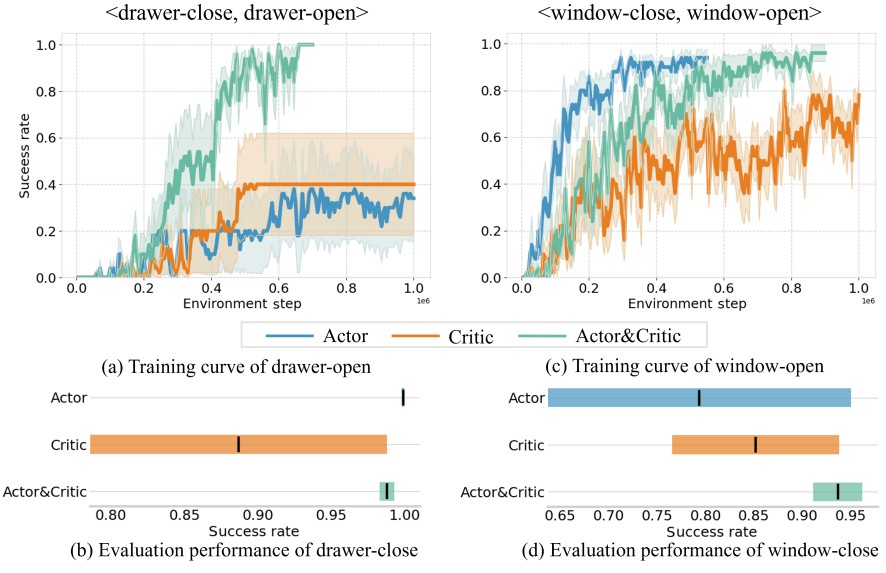

Figure 13: Additional results of NBSP variations on the Meta-World benchmark. Figures (a) and (c) illustrate the training curves for the second task, while Figures (b) and (d) show the performance on the first task after sequentially learning the task pair. (Training terminates upon stably optimal performance or maximum steps for each task.)

and retention, it still delivers robust overall performance. In terms of plasticity, NBSP performs slightly under the EWC, but its stability is considerably higher. Similarly, while NBSP's plasticity is marginally lower than that of Importance-based NBSP, its stability remains significantly better. Overall, NBSP strikes an excellent balance between stability and plasticity on the Atari benchmark, outperforming the other baselines.

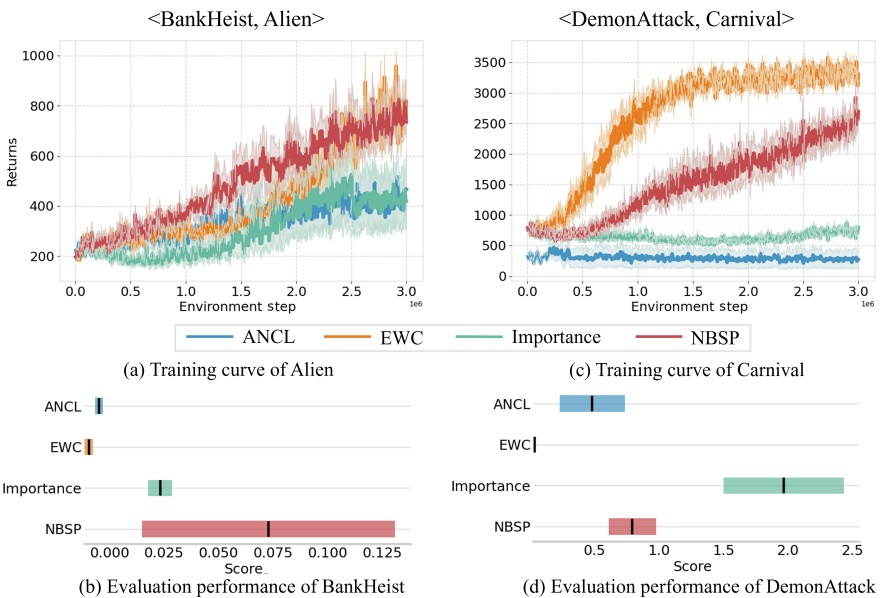

Figure 14: Additional results of NBSP on the Atari benchmark compared to other baselines. Figures (a) and (c) illustrate the training curves for the second task, while Figures (b) and (d) show the performance on the first task after sequentially learning the task pair.

To further analyze the relationship between the stability-plasticity trade-off and the number of RL skill neurons, we compare results using different quantities of RL skill neurons, as shown in Figure 15. It is evident that as the number of RL skill neurons increases, the ability to learn the new task decreases, while the retention of previously acquired knowledge improves. Therefore, adjusting the number of RL skill neurons based on specific requirements can effectively balance stability and plasticity. If stability is a priority, increasing the number of RL skill neurons is recommended. Conversely, if plasticity is more critical, reducing the number of neurons is more beneficial. This insight underscores how fine-tuning the number of RL skill neurons allows for precise control over the agent's performance in sequential tasks.

## D    ALGORITHM

The pseudo-code for SAC with NBSP is presented in Algorithm 1, which is applied to train the second task. Key differences from standard SAC are highlighted in blue. In addition to the extra input, two main modifications include the sampling process and the network update process.

## E    DISCUSSION, LIMITATION AND FUTURE WORK

**Discussion**. As shown in Section 3.1, the order in which tasks are learned significantly impacts the agent's performance. Regardless of the sequence, stability loss or plasticity loss occurs. If the agent learns a more difficult task first, it struggles to retain memory after mastering the new task. Conversely, starting with an easier task may preserve some knowledge but hinder the ability to learn subsequent tasks. Thus, task order is crucial for balancing stability and plasticity, warranting further investigation.

---

**Algorithm 1** SAC with NBSP

---

Initialize policy parameters $\theta$, Q-function parameters $\phi_1$, $\phi_2$, and target Q-function parameters $\phi'_1$, $\phi'_2$

Initialize empty replay buffer $\mathcal{D}$

Initialize replay interval $k$

**Input:** Replay buffer $\mathcal{D}_{pre}$, mask of the policy $mask_\theta$ and mask of the Q-function parameters $mask_{\phi_1}, mask_{\phi_2}$

1: **for** each iteration **do**
2:     **for** each environment step **do**
3:         Sample action $a_t \sim \pi_\theta(a_t|s_t)$
4:         Execute action $a_t$ and observe reward $r_t$ and next state $s_{t+1}$
5:         Store $(s_t, a_t, r_t, s_{t+1})$ in replay buffer $\mathcal{D}$
6:     **end for**
7:     **for** each gradient step **do**
8:         **if** step $\equiv 0 \pmod{k}$ **then** Sample batch of transitions $(s_i, a_i, r_i, s_{i+1})$ from $\mathcal{D}_{pre}$
9:         **else** Sample batch of transitions $(s_i, a_i, r_i, s_{i+1})$ from $\mathcal{D}$
10:       **end if**
11:       Compute target value:

$$y_i = r_i + \gamma \left( \min_{j=1,2} Q_{\phi'_j}(s_{i+1}, \tilde{a}_{i+1}) - \alpha \log \pi_\theta(\tilde{a}_{i+1}|s_{i+1}) \right)$$

    where $\tilde{a}_{i+1} \sim \pi_\theta(\cdot|s_{i+1})$
12:       Update Q-functions by one step of gradient descent with mask:

$$\phi_j \leftarrow \phi_j - \lambda_Q \, mask_{\phi_j} \nabla_{\phi_j} \frac{1}{N} \sum_i \left( Q_{\phi_j}(s_i, a_i) - y_i \right)^2 \quad \text{for } j = 1, 2$$

13:       Update policy by one step of gradient ascent with mask:

$$\theta \leftarrow \theta + \lambda_\pi \, mask_\theta \nabla_\theta \frac{1}{N} \sum_i \left( \alpha \log \pi_\theta(a_i|s_i) - \min_{j=1,2} Q_{\phi_j}(s_i, a_i) \right)$$

14:       Update temperature $\alpha$ by one step of gradient descent:

$$\alpha \leftarrow \alpha - \lambda_\alpha \nabla_\alpha \frac{1}{N} \sum_i \left( -\alpha \log \pi_\theta(a_i|s_i) - \alpha \bar{\mathcal{H}} \right)$$

15:       Update target Q-function parameters:

$$\phi'_j \leftarrow \tau \phi_j + (1 - \tau) \phi'_j \quad \text{for } j = 1, 2$$

16:     **end for**
17: **end for**

---

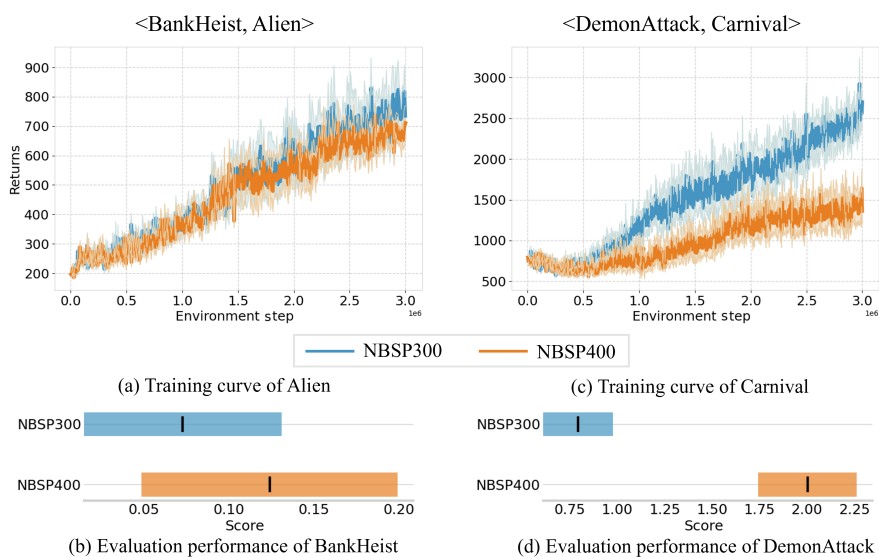

Figure 15: Trade-off between stability and plasticity in NBSP with different numbers of RL skill neurons. Figures (a) and (c) illustrate the training curves for the second task, while Figures (b) and (d) show the performance on the first task after sequentially learning the task pair.

In our NBSP approach, the number of RL skill neurons also plays a vital role in achieving this balance, as illustrated in Section C.5.5. Increasing the number of neurons enhances stability by retaining more encoded knowledge, while decreasing it promotes plasticity by allowing more neurons to focus on new knowledge. Developing an automated method to determine the optimal number of neurons is a promising avenue for future research.

**Limitation**. While the proposed NBSP method effectively balances stability and plasticity in DRL, it does have a notable limitation. Specifically, the number of RL skill neurons must be manually determined and adjusted according to the complexity of the learning task, as there is no automatic mechanism for this selection.

**Future work**. The neuron analysis introduced in this work offers a novel approach for identifying RL skill neurons, significantly enhancing the balance between stability and plasticity in DRL. The identification of RL skill neurons opens up several promising directions for future research and applications, such as: (1) Model Distillation: by focusing on RL skill neurons, it becomes possible to distill models by pruning less relevant neurons, leading to more efficient and compact models with minimal performance degradation. (2) Bias Control and Model Manipulation: RL skill neurons could be leveraged to control biases and modify model behaviors by selectively adjusting their activations. This approach could be particularly valuable in scenarios requiring specific outputs or behaviors.

Regarding to the NBSP method, its applicable potential extends beyond DRL. It could also be adapted to other learning paradigms, such as supervised and unsupervised learning, to address similar stability-plasticity challenges. In future work, we plan to explore these extensions and verify their effectiveness across various domains.

