# OpenReview forum: "Neuron-level Balance between Stability and Plasticity in Deep Reinforcement Learning"
_ICLR.cc/2025/Conference — ICLR 2025 Conference Withdrawn Submission_

### Official Review · Reviewer_6FqV · 2024-10-30

**Soundness:** 2
**Presentation:** 3
**Contribution:** 2
**Rating:** 3
**Confidence:** 4

**Summary:**

This paper introduces a method from the neuron level for mitigating plasticity loss issue while maintaining the performance in Deep RL domain.  Their method outperforms several related baselines on several simulation tasks.

About the method part, I think the total algorithm design is novel and makes sense. However, The experimental part is too brief to support the effectiveness of the algorithm.

**Strengths:**

1. This paper is well-written, and has clear figures.
2. The method is introduced in a reasonable and theatrical way.
3. The results show that their method performs well practically.

**Weaknesses:**

1.  The selected scenarios in the experimental part are too simple and rare, involving only four small tasks in metaworld, to provide a strong validity verification, which makes me doubt the performance of the method in practical applications. Please show the effectiveness of the algorithm on more benchmark tasks such as DeepMind Control Siult\Gym-Mujoco\Baby AI [1] and the famous continuous learning manipulator benchmark-- LIBERO[2].

2. The existing continuous deployment papers all build well over two sequential training tasks. Please test all methods according to the mainstream experimental design. Refer to the paper [3].

3. The selected baselines are few and simple. Please compare with the latest method of addressing plasticity loss,e.g. [4] [5] [9]. It also suggests a contrast with recent approaches to lifelong learning [8](which do not explicitly focus on plasticity loss or primacy bias, but also test an agent's performance in changing scenarios).

4. This paper claims that the proposed method can alleviate plasticity loss, but does not show the performance of the methods on plasticity evaluation metrics, such as covariance metric [6], FAU [7].

[1] Chevalier-Boisvert, Maxime, et al. "Babyai: A platform to study the sample efficiency of grounded language learning." arXiv preprint arXiv:1810.08272 (2018).

[2] LIBERO: Benchmarking Knowledge Transfer for Lifelong Robot Learning

[3]Abbas, Zaheer, et al. "Loss of plasticity in continual deep reinforcement learning." Conference on Lifelong Learning Agents. PMLR, 2023.

[4]Sokar, Ghada, et al. "The dormant neuron phenomenon in deep reinforcement learning." International Conference on Machine Learning. PMLR, 2023.

[5] Nikishin, Evgenii, et al. "Deep reinforcement learning with plasticity injection." Advances in Neural Information Processing Systems 36 (2024).

[6]  Ceron, Johan Samir Obando, Aaron Courville, and Pablo Samuel Castro. "In value-based deep reinforcement learning, a pruned network is a good network." Forty-first International Conference on Machine Learning.

[7] Ma, Guozheng, et al. "Revisiting plasticity in visual reinforcement learning: Data, modules and training stages." arXiv preprint arXiv:2310.07418 (2023).

[8] Julian, Ryan, et al. "Never stop learning: The effectiveness of fine-tuning in robotic reinforcement learning." arXiv preprint arXiv:2004.10190 (2020).

[9] Ma, Guozheng, et al. "Revisiting plasticity in visual reinforcement learning: Data, modules and training stages." arXiv preprint arXiv:2310.07418 (2023).

**Questions:**

As for the skill neuron identification part,  I think the first half is too similar to the design of $\tau$-dormant neuron [1], but it is not referenced in the method part. This kind of makes me wonder. Could you please explain this point?

[1 ]Sokar, Ghada, et al. "The dormant neuron phenomenon in deep reinforcement learning." International Conference on Machine Learning. PMLR, 2023.

---

> ### Author Response · Authors · 2024-11-25
>
> We sincerely thank the reviewer for their detailed feedback and constructive suggestions. We understand the concerns raised and will address them to strengthen the quality of our paper. Below, we provide a point-by-point response to the reviewer's comments.
>
> **Limited Experimental Scenarios and Benchmark Selection**
> We appreciate the suggestion to extend our experiments to more complex and diverse benchmarks beyond Meta-World and Atari. While we initially chose Meta-World and Atari due to their established difficulty and representativeness in reinforcement learning research, we acknowledge that the limited number of tasks tested may not fully demonstrate the generalizability of our method. In future work, we will include more challenging benchmarks such as DeepMind Control Suite, Gym-Mujoco, BabyAI [1], and the lifelong learning manipulator benchmark LIBERO [2]. These additional experiments will further validate the effectiveness of our approach in diverse and complex settings.
>
> **Sequential Task Learning and Mainstream Experimental Design**
>
> **Selection of Baselines for Comparison**
> As mentioned in our general response, we recognize the need to expand our experimental design to multi-task sequential setups and to compare against a broader range of state-of-the-art baselines, particularly those addressing plasticity loss and continual RL learning. These enhancements will provide a more comprehensive evaluation of our method and better align with mainstream practices in continual reinforcement learning research.
>
> **Evaluation Metrics for Plasticity**
> We acknowledge the need for more rigorous evaluation metrics to demonstrate our method’s impact on plasticity. In our current evaluation, we focused on stability-plasticity trade-offs but did not include specific metrics like the covariance metric [6] or the FAU metric [7]. In the revised paper, we will extend our evaluation to include these plasticity metrics, providing a more quantitative assessment of our method's effectiveness in balancing stability and plasticity.
>
> We are grateful for the additional references provided and will incorporate the relevant literature into our related work section. Here are some specific changes we plan to implement based on your feedback:
>
> - **Expanded Experiments**: We will include additional tasks from the suggested benchmarks to provide a broader evaluation of our method's applicability.
> - **Longer Task Sequences**: We will test our approach on longer sequences of tasks, aligning with the designs used in recent continual learning research.
> - **Comprehensive Baseline Comparisons**: We will include state-of-the-art baselines that address plasticity loss and lifelong learning challenges, enhancing the rigor of our comparative analysis.
> - **Incorporating Plasticity Metrics**: We will utilize the covariance and FAU metrics to provide a deeper understanding of our method’s impact on plasticity.
>
> We greatly appreciate the detailed and constructive feedback, which will guide us in enhancing the quality of our paper. Your insights have been invaluable in highlighting areas for improvement, and we are committed to making the necessary revisions to strengthen our contribution to the field. Thank you once again for your thoughtful review.

---

### Official Review · Reviewer_nKiH · 2024-11-01

**Soundness:** 2
**Presentation:** 3
**Contribution:** 3
**Rating:** 3
**Confidence:** 3

**Summary:**

This paper introduces a new method that prevents catastrophic forgetting while allowing to efficiently learning new tasks in DRL. After learning a task and during learning a second task, they propose to prevent the modification of neurons that contribute to the success of the first task. The experiments show that the proposed approach better retain the first task and better learn the second task, compared to 3 baselines on three pairs of tasks.

**Strengths:**

The paper is overall easy-to-follow and the approach is novel and positively simple. The approach tackles an important task (continual learning in DRL) with little overhead (no new parameters, no pseudo-rehearsal etc..).

**Weaknesses:**

The paper lacks small but important details. Experiments and analysis are insufficient.

**Questions:**

The experiments are not sufficient to demonstrate the generality of the method. It is currently unclear whether the results come from the specific overlap between tasks (the paper only focuses on opening/closing the same object). On Meta-world, the authors could explore new pairs of tasks based on "Push", "Reach", "pick place", "basketball", "sweep into". It was actually done for Atari in supplementary materials, but it should be added to the main paper. Best results should not be the only one showed in the main paper.

The approach is only tested on two sequential tasks. It raises the questions of how well it would perform on more sequentials tasks. Is there a limitation of the method there ? This work does not present how large is the mask ratio. It may be that they are very large such that the plasticity quickly decreases as the number of tasks increase. This should be clarified and studied. Such a clarification would start with:
"The neurons with the highest scores are identified as RL skill neurons, as they are instrumental in task-specific knowledge retention. And the number of RL skill neurons varies depending on the complexity of the task". The number of neurons is unclear. Is it a hyper-parameter ? If yes, how does it impact stability/plasticity ? A section in appendix started to investigate this question, but 1) this should be in the main paper; 2) it is unclear what the numbers "300" and "400" exactly refer to; 3) more numbers should be studied. Limitations discussed in appendix should be in the main paper.

Section 4.3 is also incomplete. The main novelty of the approach is the masking idea. But the authors do not try the method without the experience replay/with masking. It is also unclear how much of replay is dedicated to previous tasks: "Additionally, we use two separate replay buffers for experience replay: one for storing the current experiences and the other for preserving experiences from the previous task. The agent then selectively samples from these buffers to update networks". It's also unclear whether other baselines (except for the "importance" variant) use/can benefit from a similar experience replay mechanism.

The paper overlooks parts of the litterature on catastrophic forgetting, see Section 5 of [1] for instance. I understand that they did not evaluate the methods on DRL tasks, but the authors should then explain why these methods are incompatible with DRL. At least [2] tried their method with DRL and is not discussed.

Given my recommendations, the main paper seems to have a problem with the length: I would suggest: move Figure 3 to appendix. A lot of space on the left and right of plots is lost; it is likely possible to reduce the size of the plots and have 3 figures per row. Barplots could be made smaller/vertical or could be reported in a table like Table 1 (without individual success rates).

Small comments:

- line 216: The "evaluation criterion function" is unclear. Is that the reward ? How do you measure "the degree to
which the agent approaches the goal" ?
- line 60: "However, stability and plasticity attribute to the expressive capabilitie" is unclear.
- Paragraph 2, Section 3.2  is mostly redundant with the related works section
- Figure 7: why not showing the average return on the x-axis ? It would help comparing with results from a) and c).
- There are two "Fengshuo Bai, Hongming Zhang, Tianyang Tao, Zhiheng Wu, Yanna Wang, and Bo Xu. Picor:
Multi-task deep reinforcement learning with policy correction. In Proceedings of the AAAI Con-
ference on Artificial Intelligence, volume 37, pp. 6728–6736, 2023b" in the bibliography.

[1] Khetarpal, K., Riemer, M., Rish, I., & Precup, D. (2022). Towards continual reinforcement learning: A review and perspectives. Journal of Artificial Intelligence Research, 75, 1401-1476.
[2] Rusu, A. A., Rabinowitz, N. C., Desjardins, G., Soyer, H., Kirkpatrick, J., Kavukcuoglu, K., ... & Hadsell, R. (2016). Progressive neural networks. arXiv preprint arXiv:1606.04671.

---

> ### Author Response · Authors · 2024-11-25
>
> We greatly appreciate the reviewer's insightful feedback and constructive comments, which have provided valuable guidance for improving our work. Below, we address each of the reviewer's concerns and outline the changes we plan to implement in our revised paper.
>
> **Limited Task Overlap and Generality of the Method**
> We acknowledge the limitation of our current experiments, which only focus on sequential tasks involving opening/closing the same object. To demonstrate the generality of our method, we will extend our experiments on the Meta-World environment by exploring new pairs of tasks, such as "Push", "Reach", "Pick Place", "Basketball", and "Sweep Into". Additionally, as suggested, we will move some of the Atari experiments from the supplementary material to the main paper to provide a more comprehensive evaluation. We will also include additional results beyond the best-performing ones to offer a more complete picture of our method’s capabilities.
>
> **Performance on Multiple Sequential Tasks and Mask Ratio Analysis**
> We agree that testing our approach on only two sequential tasks raises questions about its scalability to longer task sequences. In the revised version, we will extend our evaluation to include learning multiple sequential tasks (e.g., 10 tasks) to assess the method's performance under more challenging settings. We will also investigate the impact of the mask ratio on stability and plasticity by varying the number of RL skill neurons, which is currently treated as a hyperparameter. This analysis will be moved from the appendix to the main paper, with a detailed study of the values "300" and "400" and their influence on learning dynamics.
>
> **Ablation Study on Gradient Masking and Experience Replay**
> We appreciate the suggestion to conduct a more thorough ablation study. In the revised paper, we will add experiments that isolate the effects of gradient masking and experience replay. Specifically, we will include:
>
> - An ablation experiment using only gradient masking without experience replay to demonstrate the individual contribution of each component.
> - An analysis of whether other baselines can benefit from the same experience replay mechanism. This will help clarify whether our method’s improvements are due to the gradient masking strategy or the replay buffers.
>
> **Additional Baselines Addressing Catastrophic Forgetting**
> We acknowledge that our current baselines may not sufficiently cover the literature on catastrophic forgetting in reinforcement learning. We will expand our experimental setup to include baselines such as Progressive Neural Networks [2] and other methods known for mitigating catastrophic forgetting [1]. We will also clarify why certain continual learning approaches are not directly applicable to DRL and highlight the unique challenges our method addresses in this context.
>
> **Paper Length and Figure Adjustments**
> Thank you for the suggestions on optimizing the paper length. We will relocate Figure 3 to the appendix and adjust the layout of existing figures to save space, for instance, by arranging plots in 3 figures per row or converting bar plots into compact tables. This will allow us to integrate more experimental results and discussions into the main paper without exceeding the page limit.
>
> **Responses to Specific Comments:**
>
> - **Evaluation Criterion Function (line 216)**
>   We appreciate the need for clarity regarding the evaluation criterion. The function varies based on the environment: in Meta-World, it is based on task success rates, while in Atari, it relies on reward scores. The degree to which the agent approaches the goal is measured using this criterion. We will clarify this in the revised text with examples for each domain.
> - **Clarification on Stability and Plasticity (line 60)**
>   The sentence will be revised for clarity: "Stability and plasticity depend on the learning capabilities of neurons. High learning capacity enhances plasticity, allowing the model to adapt to new tasks, while low capacity helps retain prior knowledge, promoting stability."
> - **Redundancy in Section 3.2**
>   We agree that Paragraph 2 of Section 3.2 overlaps with the related work section. We will streamline this content to avoid redundancy.
> - **Visualization of Average Return in Figure 7**
>   Thank you for this suggestion. We will adjust Figure 7 to include average returns on the x-axis, enabling better comparisons with other plots.
> - **Bibliography Duplication**
>   We will correct the duplication of the reference to "Fengshuo Bai et al." in the bibliography.
>
> We are grateful for your thoughtful recommendations. These revisions will significantly enhance the clarity, comprehensiveness, and impact of our paper. Thank you once again for your detailed review and suggestions.

---

> > ### Comment · Reviewer_nKiH · 2024-11-29
> >
> > I thank the authors for their answer. Since there was no revision of the paper before the deadline, I think another round of reviews with these modifications is needed. Thus, I will keep my score.

---

### Official Review · Reviewer_ho6p · 2024-11-02

**Soundness:** 1
**Presentation:** 2
**Contribution:** 1
**Rating:** 3
**Confidence:** 4

**Summary:**

This paper proposes a new continual RL framework, **Neuron-level Balance between Stability and Plasticity (NBSP)**, to address the stability-plasticity dilemma in continual deep RL. NBSP integrates three core components: (1) **RL skill neurons**, (2) **gradient masking**, and (3) **experience replay**. The authors introduce a goal-oriented method to identify and quantify RL skill neurons, using a ranking and thresholding approach similar to dormant neurons. To mitigate forgetting, NBSP applies a gradient mask to parameters connected to those  skill neurons at the output side, preserving the critical parameters learned from the first task from alteration. Additionally, NBSP periodically samples experience from previous tasks as a rehearsal strategy to reinforce memory retention. The framework is evaluated on sequential task setups from Meta-World and Atari benchmarks, each consisting of two tasks.

**Strengths:**

- The paper addresses an important and relatively underexplored challenge in reinforcement learning by examining ways to balance stability and plasticity in a continual learning setting. The topic aligns well with the conference's core themes.

- The paper is clearly structured and well-organized, making it easy to follow and understand.

- Overall, the idea of addressing plasticity-stability trade-off at the neuron-level through identifying the RL skill neurons is interesting. The use of both neuron activation and goal-oriented behavior measures in the scoring function is somewhat novel. Neuron-level algorithms indeed present promising directions for future research in continual reinforcement learning.

**Weaknesses:**

1) **Comprehensive Review of Existing Approaches in Continual RL**: The paper lacks a comprehensive review of existing approaches in continual reinforcement learning. Given that stability-plasticity trade-off is a common challenge in continual RL, a survey of how it has been addressed would provide valuable context. In the fist paragraph of the related works section, the authors focus primarily on discussing continual learning without specific reference to continual RL. Additionally, for neuron-level research, a discussion of neuron-level continual RL methods, especially the **structure-based continual RL approaches** [1][2][3], would be helpful, as these methods also address the problem at the neuron level and a clear discussion is essential.  Since the experience replay is also highlighted as part of the paper's contribution, the related work also needs to discuss with **rehearsal-based continual RL methods** (e.g., [4][5]).

   [1] Using task descriptions in lifelong machine learning for improved performance and zero-shot transfer. *JAIR 2020*.

   [2] Continual Task Allocation in Meta-Policy Network via Sparse Prompting. *ICML 2023*.

   [3] Packnet: Adding multiple tasks to a single network by iterative pruning. *CVPR 2018*.

   [4] Efficient Lifelong Learning with A-GEM*. *ICLR 2021*.

   [5] Disentangling Transfer in Continual Reinforcement Learning. *Neurips 2022*.


2) **Clear Definition of the Continual Learning Problem**: The paper lacks a clear and precise definition of the continual learning problem it aims to address in the main body of the paper. Important aspects, such as whether the method is task-incremental or class-incremental, the extent of access to previous task data (and any limitations on this), and whether the continual learning model involves only the policy network or both the policy and critic networks, are not specified. Also, a clear definition to the "RL skill neuron" is required to clarify the concept.

3) **Novelty of Combining Task-Specific Skill Neurons with Gradient Masking**: The idea of combining task-specific skill neurons with gradient masking is not new. Similar approaches have been explored in previous works, such as PackNet [3] and CoTASP [2]. PackNet, prunes the policy after training each task to identify the most important neurons, which are similar to the “skill neurons” proposed here, while CoTASP combines pre-allocation and adaptive updates on "skill neuron selection masks". Both approaches use gradient masking to protect task-specific neurons and have shown strong capability to handle more complex continual RL tasks without the need of network inflation or data rehearsal from previous tasks. A thorough discussion and comparison with these closely related methods would help clarify the specific contributions and advantages of the proposed approach.

4) **Limitations in the Score Function**: The score function for identifying skill neurons is not very convincing.  It compares neuron activation  $a(N,t)$  and reward criterion $q(t)$  with some simple baselines, and then applies an indicator function to filter neurons with above-average activation and reward:

   4.1) For the activation component, the authors seem to use raw activation values instead of  **absolute** value;

   4.2) The activation baseline $\bar{a}(N)$ is calculated as a mean over multiple time steps for **a single neuron** rather than across neurons, though its meaning is to distinguish activation pattern for neurons from one to another;

   4.3) The indicator function in Eq 3 results in binary **counts of active steps**, losing precision in distinguishing the quality of activations (e.g., when two steps both meet the condition, it does not capture how much one might exceed the threshold over another);

   4.4) Since $q(t)$ is derived from some reward signal, all neurons from the networks will receive the same score over the same period, making it less effective as a **neuron-level measure**;

   4.5) The intuition behind Eq (4) is unclear. It assumes that **neurons not meeting Eq (3) still contributes positively** assigning them with a score of $1-Acc(N)$. This makes the intuition behind the scoring mechanism for RL skill neurons quite confusing. It's unclear what kind of neurons are eventually selected as skill neurons.  A statistical analysis of this scoring mechanism is needed to clearly illustrate the intuition behind this score function.

5) **Similarity to Dormant Neurons**: The way to score RL skill neurons closely resembles the dormant neuron approach in the literature. It would be helpful to explain why dormant neurons were not directly applied and to include a comparison with a dormant-based scoring function in the experiments.

6) **Gradient Masking**: The neuron-level identification strategy is transformed to parameter-level protection strategy by blocking all gradients connected to the skill neuron, even though each parameter links two neurons from consequent layers. Why not consider blocking $\triangle W_{j,:}$ too, or blocking $\triangle W_{i,j}$ with both sides being skill neurons? The assumptions behind this needs to be clearly stated.

7) **Gradient Marking with Experience Replay**: Experience replay inherently requires access to previous task data, which may be restricted in certain continual learning scenarios. The approach would be more impactful if gradient masking could achieve effective performance without the need for experience replay.

8) **Implementation for the Skill Neurons**: The paper lacks detailed discussion on the implementation strategy and insights for the proposed score function. It appears that RL skill neurons are only identified once at the end of training the first task. For scenarios with more than two tasks, there is no comments on how the masks accumulate across tasks. The authors should not restrict to the two-task settings, and consider increasing the scalability of the proposed approach.

9) **Simple Finetuning Baseline**: The illustration of stability and plasticity challenges in Section 3.1 uses a two-task fine-tuning baseline, which lacks continual learning techniques and is outdated. The authors should provide a stronger foundation by using some decent continual RL approaches for motivating the key challenges and showcasing stability-plasticity trade-offs.

10) **Limitations of Experimental Domain**: The experimental domain, limited to two simple tasks, is overly simple. Meta-World is commonly used in continual RL, and Continual World (CW) provides a more challenging benchmark with open-source baselines, comprehensive evaluation criterias, and performance curves. The Meta-World “window-close” task adopted in this paper is also part of CW, where current methods could achieve near 100% success without replay. I recommend evaluating the on standard CW10 or CW20 benchmarks and comparing with state-of-the-art methods, such as neuron-level baseline CoTASP, PackNet and rehearsal baseline ClonEx-SAC.


11) **Ablation Study Suggestions**:  The ablation study would benefit from incorporating some of the following baselines: (1) alternative activation functions; (2) Eq 4 with Acc(N) only; (3) using dormant neuron score function. (4) some randomly selected neurons as Skill Neurons; (5) gradient-masking only, without replay; (6) gradient masking with $\triangle W_{i,j}$

**Questions:**

- Why the evaluation criteria $q_\theta(t)$ is made binary for Meta-World and is based on return for Atari? Meta-World provides step-wise dense rewards that offer a progressive measure of the agent’s goal accomplishment, which would make it more informative for evaluation.

- How are reasonable time steps T determined in Eq (1) and Eq (2) for different continual RL environments?

- How does the reward-based evaluation criterion relate to the identification of neuron-level RL skill neurons? More explanation and statistical insights on this relationship will be helpful.

- Why does the method use raw activation rather than absolute activation?

- Is the proposed method specific to certain activation functions, such as ReLU, and does it perform differently with alternatives like tanh?

- The experience replay frequency is quite high. How much data must be stored from previous tasks to support rehearsal?

- A minor comment on learning curves presented: throughout the paper, training curves are shown without clear indication of whether they correspond to the result when training the first or second task. Including full training curves would make it easier for readers to interpret results.

- In practice, how do you sample data and compute q(t) from the experience replay?

- Could you provide computational complexity for measuring the RL skill neuron?

- The algorithm 1 provided in appendix lacks the crucial steps of computing RL skill neurons. Could you reflect this procedure more precisely?

**Details Of Ethics Concerns:**

N.A.

---

> ### Author Response · Authors · 2024-11-25
>
> We sincerely appreciate the reviewer's detailed feedback and valuable suggestions, which have greatly helped us identify areas for improvement in our work. Below, we address each of your concerns and outline the modifications we plan to make in our revised paper.
>
> 1. **Comprehensive Review of Existing Approaches in Continual RL**
>    We acknowledge that our related work section primarily focused on general continual learning and did not sufficiently address continual RL methods, especially in the context of stability-plasticity trade-offs. In the revised version, we will include a comprehensive survey of continual RL, particularly focusing on structure-based methods \[1\]\[2\]\[3\] and rehearsal-based approaches \[4\]\[5\], as suggested. This will provide better context and positioning for our work. Additionally, we will expand the discussion to include neuron-level methods to highlight the relevance of our approach.
> 2. **Clear Definition of the Continual Learning Problem**
>    We agree that the definition of the continual learning setting in our paper was not sufficiently clear. We focused on a two-task setup, which led to ambiguity regarding whether the approach is task-incremental or class-incremental. In the revised paper, we will explicitly define the continual RL problem, detailing whether the method applies to both the policy and critic networks, and clarifying the extent of access to previous task data. Furthermore, we will include a precise definition of the "RL skill neuron" to enhance clarity.
> 3. **Novelty of Combining Task-Specific Skill Neurons with Gradient Masking**
>    We understand the reviewer's concern regarding the novelty of combining task-specific skill neurons with gradient masking, as similar ideas were explored in PackNet [3] and CoTASP [2]. However, our method identifies RL skill neurons driven by reinforcement learning objectives, specifically focusing on neurons that contribute to the policy's performance. This distinction allows us to retain RL-specific knowledge more effectively. In the revision, we will include a thorough comparison with these methods in both the related work and experimental sections to highlight our contributions more clearly.
> 4. **Limitations in the Score Function**
>    We acknowledge the reviewer's concerns regarding our score function:
>    - **(4.1) Raw Activation Values**: The implementation indeed uses absolute activation values, but this was not clearly explained. We will clarify this in the paper.
>    - **(4.2) Activation Baseline**: We compare each neuron's activation with its own average over time, rather than across neurons, to capture its relative importance.  The choice to compare each neuron's activation with its own average over time is inspired by practices in neuroscience, where neural activity is often evaluated relative to its baseline firing rate to highlight deviations that indicate task-specific importance. This neuron-specific baseline allows us to focus on each neuron's unique contribution, avoiding the noise and dilution that can arise when averaging across neurons with varying roles. By capturing these individual deviations, our method effectively identifies neurons that are particularly relevant for skill retention, ensuring a more precise assessment of their functional importance.
>    - **(4.3) Binary Counts in Eq (3)**: We appreciate the suggestion to consider finer distinctions in activation levels. In future work, we will explore more granular measures to enhance precision.
>    - **(4.4) Reward Signal in q(t)**: Our intention was to use q(t) as a global measure of goal achievement. We will improve the explanation to avoid confusion.
>    - **(4.5) Intuition Behind Eq (4)**: The scoring mechanism aimed to consider both high and low activation scenarios. We will refine our explanation and include a statistical analysis of the score distribution in the experiments.
> 5. **Similarity to Dormant Neurons**
>    The method for identifying RL skill neurons was designed to consider both activation levels and task performance, unlike dormant neurons that rely solely on activation thresholds. Continual RL tasks are inherently goal-driven, with each task demanding specific behaviors and policies to optimize performance. Simply relying on activation thresholds might fail to capture neurons that are critical for task-specific goals.By integrating both activation levels and task performance into our scoring function, we ensure that the selection process aligns with the task's objectives. This dual consideration allows us to identify neurons that not only exhibit consistent activity but also contribute meaningfully to achieving task-specific goals. To demonstrate the strengthen of our method, we will include a comparison with dormant neuron-based methods in the experiments to demonstrate the benefits of our approach.

---

> ### Author Response · Authors · 2024-11-25
>
> 6. **Gradient Masking Strategy**
>    We appreciate the reviewer's suggestion to explore alternative gradient masking strategies. Currently, we mask gradients connected to identified skill neurons based on their output dimensions. We plan to experiment with additional strategies, such as blocking gradients between specific neurons across layers, in future work.
> 7. **Gradient Masking with Experience Replay**
>    We agree that relying on experience replay may limit the applicability of our approach in certain settings. However, we observed that gradient masking alone led to some knowledge forgetting, which was mitigated by incorporating experience replay from the training buffer. In the revised paper, we will discuss potential methods to reduce dependency on replay data and explore alternative rehearsal strategies.
> 8. **Implementation for the Skill Neurons**
> 9. **Simple Fine-tuning Baseline**
> 10. **Limitations of Experimental Domain**
>    Regarding the implementation of skill neurons, the selection of baselines, and the experimental domain, we refer to our general response for a detailed discussion of our planned improvements. Briefly, we acknowledge the need to provide more detailed descriptions of our methodology, adopt advanced continual RL baselines, and expand our experiments to more challenging multi-task settings such as Meta-World's CW10 and CW20. These enhancements aim to address the limitations identified and further validate the robustness and applicability of our approach.
> 11. **Ablation Study Suggestions**
>    We appreciate the suggestions for a more detailed ablation study. In addition to our current analysis, we will include comparisons with: (1) alternative activation functions; (2) the score function using Acc(N) only; (3) dormant neuron-based scoring; (4) randomly selected skill neurons; and (5) gradient masking without replay.
>
> **Responses to Specific Questions:**
>
> - **Evaluation Criteria (qθ(t))**: We chose different evaluation criteria based on task-specific objectives. For Meta-World, we used success rates, aligning with standard evaluations for these tasks.
> - **Time Steps (T) in Eq (1) and Eq (2)**: This is a hyperparameter set to 1e6 for all experiments. We agree that this choice should be more adaptive and will explore fine-tuning it for different environments.
> - **Reward-based Evaluation and Skill Neurons**: If neuron activation and reward performance are positively correlated, we classify it as an RL skill neuron. We will add graphical representations to highlight these correlations.
> - **Use of Raw vs. Absolute Activation**: Thanks for the valuable suggestions, we do use absolute activation values and we revise the expression to absolute activation values in paper for better clarity.
> - **Impact of Activation Functions**: The current approach uses tanh in SAC networks. We will explore other activation functions in future studies to verify the generality of our method.
> - **Experience Replay Frequency**: We retain the replay buffer of size 1e6 for the previous task. During training, we only use this stored data.
> - **Learning Curves Clarification**: The upper part of each figure represents the first task, while the lower part corresponds to the second task. We will adjust the figures for better clarity.
> - **Sampling and q(t) Computation**: The evaluation criteria q(t) are directly obtained from interactions with the environment, not from replay buffers.
> - **Computational Complexity**: The identification of RL skill neurons involves additional evaluation steps after training the first task, with complexity similar to the RL algorithm itself.
> - **Algorithm 1 Enhancement**: We will revise Algorithm 1 in the appendix to include the steps for computing RL skill neurons.
>
> We are grateful for your constructive feedback and believe these revisions will significantly improve the clarity and impact of our work. Thank you for your insightful suggestions.

---

### Official Review · Reviewer_fB6r · 2024-11-04

**Soundness:** 3
**Presentation:** 3
**Contribution:** 3
**Rating:** 3
**Confidence:** 4

**Summary:**

The paper makes three main contributions: (1) it introduces the concept of RL skill neurons, a novel approach specifically tailored to deep reinforcement learning, which identifies neurons crucial for retaining task-specific knowledge; (2) it proposes the Neuron-level Balance between Stability and Plasticity (NBSP) framework, utilizing gradient masking to balance stability and plasticity at the neuron level; and (3) it provides experimental validation on the Meta-World and Atari benchmarks, demonstrating that NBSP effectively preserves prior knowledge while adapting to new tasks.

**Strengths:**

1. The paper is well-written, presenting complex ideas clearly and understandably.
2. It introduces a novel method for identifying "activated" neurons, contributing a fresh perspective on neuron-level balancing between stability and plasticity in deep reinforcement learning. This approach, which targets specific neurons for skill retention and adaptability, is a noteworthy advancement in handling the stability-plasticity dilemma.

**Weaknesses:**

1. **Limited Experimental Scope and Scalability Concerns**:
    - While the method presents a promising approach to continual learning, the experimental setup explores only two task sequences, which limits insights into scalability. As the number of tasks in a sequence grows, tracking neurons specific to each task may pose scalability issues. It would be insightful to extend the experiments to a sequence of approximately 10 tasks to demonstrate the method’s scalability and robustness in long-term continual learning settings.
2. **Restricted Baseline Comparisons**:
    - The paper lacks comparisons with key baselines from reinforcement learning literature. Methods like [1], [2], and [3], commonly used in reinforcement learning, could provide a stronger basis for evaluating the relative effectiveness of the proposed approach. Including such baselines would help contextualize the results and address any gaps in performance evaluation against established methods.

[1] Loss of Plasticity in Deep Continual Learning., Dohare et al., Nature 2024

[2] Prediction and Control in Continual Reinforcement Learning., Anand et al., NeurIPS 2023

[3] Loss of Plasticity Continual Deep Reinforcement Learning., Abbas et al., CoLLAs 2023

**Questions:**

Please refer to the Weakness

---

> ### Author Response · Authors · 2024-11-25
>
> Thank you for your constructive feedback and for highlighting the key areas where our work can be further strengthened. We appreciate the opportunity to clarify and improve our contributions in response to your insightful comments.
>
> 1. **Limited Experimental Scope and Scalability Concerns**:
>
>    We acknowledge your concern regarding the scalability of our method, especially in the context of long-term continual learning settings. The current experiments were indeed limited to two task sequences, which may not fully capture the scalability and robustness of our approach. In response, we have planned additional experiments with a sequence of approximately 10 tasks to evaluate the scalability and effectiveness of our method in more complex and extended scenarios. These results will be included in the revised version to provide a deeper understanding of the method’s performance across a broader range of tasks.
>
> 2. **Restricted Baseline Comparisons**:
>
>    We appreciate your suggestion to include comparisons with more diverse and established baselines from the reinforcement learning literature. In the current version, we focused on addressing the stability-plasticity dilemma by comparing our approach with three key baselines. However, we understand the importance of benchmarking against additional methods commonly used in continual reinforcement learning. We are in the process of incorporating comparisons with the methods you suggested ([1], [2], and [3]), as well as other state-of-the-art baselines. This will allow us to present a more comprehensive evaluation of our approach's performance relative to existing techniques.
>
> We are confident that these enhancements will address the concerns raised and further demonstrate the strengths and scalability of our proposed method.
>
> Thank you once again for your valuable comments.

---

### Author Response · Authors · 2024-11-25
**General Response**

We sincerely thank all the reviewers for their thoughtful feedback and valuable suggestions. Your constructive comments and recognition of our contributions have provided great encouragement and clear guidance for improving the paper. Below, we summarize the strengths of our work as highlighted by the reviewers and outline our plans for addressing the points raised.

### **Positive aspect**

We are grad that the reviewers have recognized our work, which greatly motivates us. All reviewers highlighted the novelty of our approach, particularly our neuron-level method for addressing the stability-plasticity dilemma in continual reinforcement learning. By identifying "activated" RL skill neurons, our method offers a fresh perspective on balancing skill retention and adaptability, which was consistently acknowledged as an innovative and impactful contribution. Additionally, the clarity and organization of the paper, along with its structured explanations and effective visuals, were widely commended, underscoring the accessibility and rigor of our presentation. The simplicity and efficiency of our approach, avoiding additional parameters or complex mechanisms like pseudo-rehearsal, further affirm its practical relevance and potential for broad applicability.

### **Commitment to Improvements**

We are grateful for the constructive feedback and detailed suggestions provided by the reviewers. We will implement the following revisions in the future version of the paper to address the concerns raised:

1. **Experimental Design**
   We acknowledge the reviewers' concerns regarding the simplicity of our experimental setup, which currently involves two tasks. This design choice was inspired by prior works [1] [2]. The two-task setup offers a streamlined framework to focus on the stability-plasticity trade-offs during task transitions. This allows for precise evaluation of key components like the Actor and Critic networks in the SAC (Soft Actor-Critic) algorithm without the added complexity of multi-task learning.

   We also recognize the importance of addressing scalability concerns. The insights gained from this setup serve as a foundation for tackling more complex multi-task and long-term continual learning scenarios. In future work, we plan to extend our experiments to include continuous multi-task settings to further validate the effectiveness of our approach in diverse and challenging environments.

2. **Baseline Comparisons**
   While our current study includes comparisons with three representative baselines addressing the stability-plasticity dilemma, we appreciate the reviewers' suggestion to expand our evaluation. Future iterations will include additional state-of-the-art baselines, particularly those highlighted by the reviewers, as well as other relevant approaches in continual reinforcement learning. This comprehensive comparison will help strengthen the robustness and reliability of our work.

By addressing these aspects, we aim to further enhance the rigor and applicability of our work, aligning it more closely with the expectations of the continual learning research community.

[1] Goodfellow I J, Mirza M, Xiao D, et al. An empirical investigation of catastrophic forgetting in gradient-based neural networks[J]. arXiv preprint arXiv:1312.6211, 2013.

[2] Wolczyk M, Zając M, Pascanu R, et al. Disentangling transfer in continual reinforcement learning[J]. Advances in Neural Information Processing Systems, 2022, 35: 6304-6317.

---

### Note · Authors · 2025-01-25

I have read and agree with the venue's withdrawal policy on behalf of myself and my co-authors.